# Functional analysis of the human perivascular subarachnoid space

Per Kristian Eide [1,2] ✉ & Geir Ringstad [3,4]

The human subarachnoid space harbors the cerebrospinal fluid, which flows within a landscape of blood vessels and trabeculae. Functional implications of subarachnoid space anatomy remain far less understood. This study of 75 patients utilizes a cerebrospinal fluid tracer (gadobutrol) and consecutive magnetic resonance imaging to investigate features of early (i.e. within 2-3 h after injection) tracer propagation within the subarachnoid space. There is a time-dependent perivascular pattern of enrichment antegrade along the major cerebral artery trunks; the anterior-, middle-, and posterior cerebral arteries. The correlation between time of first enrichment around arteries and early enrichment in nearby cerebral cortex is significant. These observations suggest the existence of a compartmentalized subarachnoid space, where perivascular ensheathment of arteries facilitates antegrade tracer passage towards brain tissue. Periarterial transport is impaired in subjects with reduced intracranial pressure-volume reserve capacity and in idiopathic normal pressure hydrocephalus patients who also show increased perivascular space size.

For centuries, it was known that the brain is covered by three meninges: the dura mater, the arachnoid mater, and the pia mater[1]. Since the discovery in 2015 of meningeal lymphatic vessels capable of draining substances from the cerebrospinal fluid to the extracranial lymph nodes[2,3], there has been renewed interest in the function of brain meninges, particularly in regards to lymphatic drainage of toxic products from brain metabolism (e.g. amyloid-β and tau peptides)[4], and immunological activity[5–7]. More recently, a fourth meningeal layer, known as the subarachnoid lymphatic-like membrane (SLYM), was described in rodents[8], and proposed to segregate the subarachnoid space into an outer and inner layer. Also human studies have focused on the anatomic organization of the subarachnoid space, which is further known to consist of arachnoid trabeculae and membranes, creating distinct regions or cisterns within the space[9]. However, the impact on cerebrospinal fluid (CSF) flow exerted by these anatomical boundaries remains poorly understood. In general, outdated conceptions of CSF flow physiology are presently under substantial revision, possibly having large implications for how we understand the impact of CSF flow on brain clearance and CSF as a

carrier of solutes and cells participating in neuroimmune cross-talk at the meninges[10].

In this study, we investigated features of early (i.e. within first 2–3 h) CSF tracer propagation within the human subarachnoid space at magnetic resonance imaging (MRI). By assessing the time-dependent movements of tracer, we sought to gain functional information about CSF dynamics within the subarachnoid space and its correlations with tracer enhancement in brain as well as measures of intracranial pressure (ICP). Here, we studied the enrichment of the intrathecal tracer in subarachnoid spaces within 2–3 h after its administration, while in the previous studies, we explored how an intrathecal tracer enriches the brain[11], subarachnoid space[12] and parasagittal dura[13] after several hours, particularly focusing on peak tracer enrichment after about 24 h.

The present findings suggest that directional perivascular (i.e. periarterial) transport within the subarachnoid space facilitates enrichment of CSF and intrathecal drugs in brain tissue. This transport was shown impaired by abnormal intracranial pulsations at ICP monitoring, indicative of impaired intracranial pressure-volume reserve capacity. Moreover, the dementia subtype idiopathic normal pressure

[1]Department of Neurosurgery, Oslo University Hospital - Rikshospitalet, Pb 4950 Nydalen, N-0424 Oslo, Norway. [2]KG Jebsen Centre for Brain Fluid Research, Institute of Clinical Medicine, Faculty of Medicine, University of Oslo, PB 1072 Blindern, N-0316 Oslo, Norway. [3]Department of Radiology, Oslo University Hospital- Rikshospitalet, Pb 4950 Nydalen, N-0424 Oslo, Norway. [4]Department of Geriatrics and Internal medicine, Sorlandet Hospital, 4838 Arendal, Arendal, Norway. ✉e-mail: p.k.eide@medisin.uio.no

hydrocephalus (iNPH) showed enlarged perivascular subarachnoid spaces and impaired periarterial tracer propagation.

## Results

### In vivo tracer evidence for a perivascular compartmentalizing of the human subarachnoid space

We first investigated features of early CSF tracer propagation within the human subarachnoid space. For this purpose, we obtained time-series of CSF tracer-enhanced MRI in a cohort of 75 subjects (Table 1; Source Data file). After administration of the tracer (gadobutrol, 0,5 mmol) in CSF at the lumbar level, the spinal transit time of the tracer to the foramen magnum was 13.8 ± 6.3 min.

Intracranially, the tracer distributed freely within the subarachnoid basal cisterns with no signs of barriers or compartmentalization. Exterior to the gyrencephalic brain surface, the tracer consistently propagated in sulci inhabited by the large artery trunks and in an antegrade (downstream) fashion. Furthermore, diffuse tracer enhancement within the subarachnoid space was typically preceded by a rim of tracer enrichment surrounding the arteries residing within the subarachnoid space. In image slices perpendicular to the artery orientation, the initial tracer enrichment formed a donut-shaped covering around the vessel wall (Fig. 1a–d). This circumferential tracer enrichment was temporary and subsequently followed by tracer enrichment within the surrounding subarachnoid space, typically at the next or the following imaging sequence.

The pattern of periarterial tracer enrichment was consistently observed in at least one or several locations in the presently examined subjects (Table 1). Perivascular enhancement within the subarachnoid space is shown for the anterior cerebral artery (ACA; Fig. 2a–d), middle cerebral artery (MCA; Fig. 2e–h), and posterior cerebral artery (PCA; Fig. 2i–l). Further examples of perivascular tracer enrichment are given for ACA (Supplementary Fig. 1), MCA (Supplementary Fig. 2; Supplementary Movie 1) and PCA (Supplementary Fig. 3).

Since perivascular tracer enrichment was typical for the large artery trunks ACA, MCA and PCA, we further asked for in vivo evidence of a perivenous subarachnoid space. As shown in Supplementary Fig. 4, a perivenous pattern of tracer enhancement was occasionally detected. However, veins at the brain surface are generally harder to depict at MRI, and thus also perivenous enhancement. We identified a perivenous tracer enrichment in very few instances only, and in those cases always in conjunction with periarterial enhancement. Therefore, the dataset is highly suggestive that the foremost front of perivascular tracer enhancement in subarachnoid space is primarily periarterial, not perivenous. We found no cases where a local crossing phenomenon of arteries and veins was considered to explain occasional perivenous enhancement. Occasional perivenous enhancement was typically seen in conjunction with the appearance of a more diffuse tracer enhancement in subarachnoid space outside the perivascular subarachnoid space. Therefore, the evidence principally points towards a periarterial propagation of solutes in subarachnoid space, not perivenous.

Taken together, these observations indicate a periarterial sheath that compartmentalizes the human subarachnoid space. Here, this perivascular compartment created a functional barrier for the tracer within the subarachnoid space.

### The periarterial subarachnoid space facilitates tracer transport

We then assessed the timing of when periarterial enhancement first occurred at different levels of the ACA and MCA. Distal perivascular

## Table 1 | Patient material

| | |
|---|---|
| **N** | 75 |
| **Age (years)** | 49.5 ± 18.8 |
| **Sex (Female/Male)** | 47/28 |
| **Body mass index (kg/m²)** | 26.7 ± 4.9 |
| **[a]Spinal transit time of intrathecal tracer (min)** | 13.8 ± 6.3 |
| **Vessels showing periarterial CSF tracer enrichment (N, %)** | |
| Anterior cerebral artery (ACA) | 66 (89%) |
| Medial cerebral artery (MCA) | 69 (93%) |
| Posterior cerebral artery (PCA) | 52 (70%) |
| Vertebral artery (VA) / Basilar artery (BA) | 0 |
| **First-time appearance of periarterial tracer (min)** | |
| **ACA** | |
| A1 (n = 11) | 23.1 ± 10.4 |
| A2 (n = 61) | 48.3 ± 46.1 |
| Pericallosal artery (n = 63) | 82.2 ± 60.3 |
| **MCA** | |
| M1 (n = 24) | 37.8 ± 47.0 |
| M2 (n = 68) | 53.1 ± 50.5 |
| M3 (n = 59) | 82.6 ± 61.5 |
| **[b]Tracer enrichment in gray matter at 2 h after i.th. injection (%)** | |
| Frontal cortex (n = 75) | 19.2 ± 19.4 |
| Temporal cortex (n = 75) | 26.1 ± 24.2 |

Continuous data given as mean ± standard deviation. [a]Spinal transit time refers to time from intrathecal injection of tracer until first appearance of tracer in cisterna magna. [b]Tracer enrichment refers to the percentage change in normalized T1 signal units 2 h after intrathecal injection.

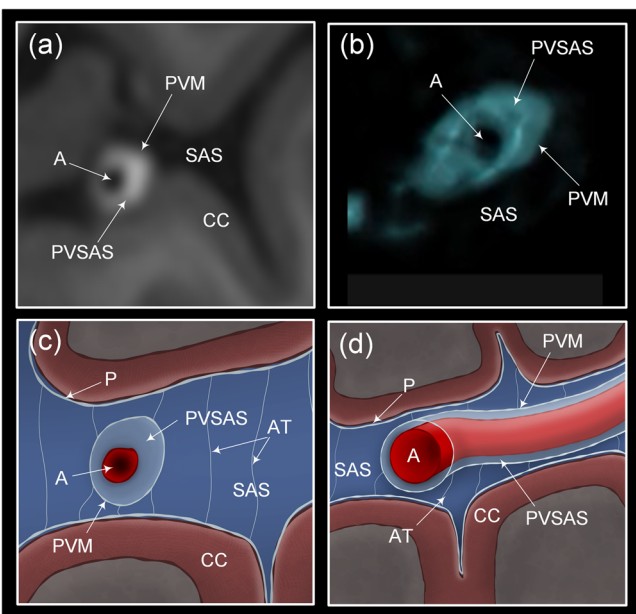

**Fig. 1 | The human subarachnoid space is compartmentalized by a perivascular subarachnoid space.** We used an MRI contrast agent (gadobutrol) as CSF tracer to study compartmentalization of the subarachnoid space (SAS). **a**, **b** In MR image planes orthogonal to the vessels, the CSF tracer that was administered intrathecal formed a donut-shaped form around the arteries (A). This perivascular subarachnoid space (PVSAS) is thus represented by the contrast-enriched perivascular compartment, delineated by a perivascular membrane (PVM) semipermeable to the CSF tracer. Tracer enrichment in PVSAS preceded tracer enrichment in surrounding subarachnoid space (SAS) and thereafter in cerebral cortex (CC). In **b** is shown a 3D representation of the PVSAS residing within the SAS. **c**, **d** Schematic illustrations show the artery (A), perivascular subarachnoid space (PVSAS), delineated by the perivascular membrane (PVM), and surrounding SAS. Provided the PVM is part of the leptomeninges (arachnoid and pia), we may anticipate that the perivascular membrane (PVM) is attached to the arachnoid trabecula (AT) and further towards the pia mater (P) and the arachnoid barrier cell layer towards the dura mater (not shown here). Illustration in **c**, **d**: Øystein Horgmo, University of Oslo.

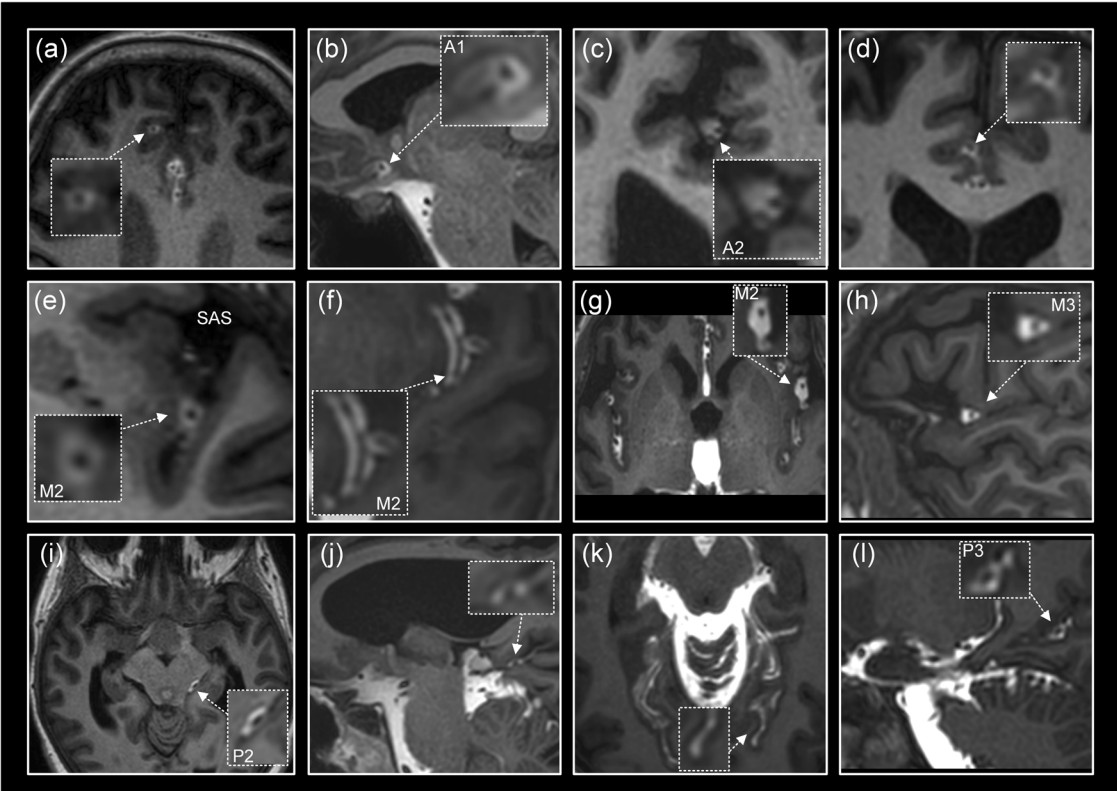

**Fig. 2 | Tracer evidence of a perivascular subarachnoid space along the anterior, middle and posterior cerebral arteries.** First signs of antegrade tracer enhancement along artery trunks in subarachnoid space (SAS) were circumferential around the anterior cerebral artery (ACA; **a–d**), middle cerebral artery (MCA; **e–h**) and posterior cerebral artery (PCA; **i–l**), where tracer enrichment in the surrounding SAS occurred subsequently at later time points. In image planes orthogonal to the vessels, tracer formed a donut-shaped form around the arteries **a–e**, **g–h**. The perivascular subarachnoid space is thus represented by the contrast-enriched perivascular compartment. Time from intrathecal tracer injection: (**a**) 199 min, (**b**) 32 min, (**c**) 60 min, (**d**) 39 min, (**e**) 53 min, (**f**) 19 min, (**g**) 10 min, (**h**) 37 min, (**i**) 20 min (**j**) 11 min, (**k**) 59 min, and (**l**) 9 min.

tracer enrichment occurred later than proximal perivascular tracer enrichment (Table 1). The time series of perivascular tracer propagation showed time-dependent enrichment along arteries in the distal direction; Fig. 3a–e shows time-dependent tracer enrichment along the ACA. Accordingly, first-time perivascular appearance of tracer occurred later the more distal the registration was along the arterial three of ACA (Fig. 3f). This is further shown in a 3D image (Fig. 3g). The perivascular tracer enrichment and subsequent tracer enrichment in subarachnoid space was later followed by enrichment in brain tissue (see signal change in brain parenchyma in Fig. 3d, e), showing that tracer enrichment in brain is also dependent on preceding enrichment in the subarachnoid space. These observations suggest that the periarterial route may facilitate directional tracer transport within the subarachnoid space antegrade along arteries, as illustrated in Fig. 3h. Time-dependent tracer enrichment in the perivascular subarachnoid space of MCA is further shown in Fig. 4a–e. As shown in Fig. 4f, tracer enrichment along M2 after 2 h was significantly stronger in perivascular subarachnoid space as compared with surrounding subarachnoid space. At later time point (about 3 h), this difference became non-significant due to the passage of tracer to the subarachnoid space. The phenomenon that periarterial enhancement of tracer occurs first in its passage towards the periphery, indicate a perivascular subarachnoid space surrounded by a membrane with a barrier function, and that periarterial transport is faster than transport in the outer subarachnoid space. The equalization of enhancement between perivascular subarachnoid space and outer subarachnoid space at later time points may indicate spread of the tracer from perivascular subarachnoid space to outer subarachnoid space through a semipermeable barrier ensheathing the perivascular subarachnoid space,

or that enhancement of outer subarachnoid space occurs in parallel, but later. Figure 4g shows how tracer enrichment in perivascular subarachnoid space was stronger than in surrounding subarachnoid space. An example of the antegrade tracer enrichment towards the distal direction along PCA is shown in Supplementary Fig. 5. The spinal transit time was no confounder of the differences in first-time appearance of tracer along the ACA or MCA.

There was a considerable inter-individual variation regarding rate of antegrade perivascular transport, measured as first-time appearance of tracer along the A2 and pericallosal artery branches of ACA, and along the M2 and M3 artery branches of MCA (Supplementary Fig. 6).

## The perivascular subarachnoid space communicates directly with the basal cisterns

At which level does an intrathecal tracer enter the periarterial subarachnoid space? The present data suggest that the intrathecal tracer enriches this space peripheral to the basal cisterns underneath the surface of the cerebral hemispheres. According to our observations, the tracer seemed to have an unrestricted pathway within the basal cisterns towards the perivascular subarachnoid space of the major artery trunks at the surface of the gyrencephalic cerebral hemispheres (Fig. 5). In line with this assumption, from visual inspection, we found that the tracer in the posterior fossa and basal cisterns enriched simultaneously with early enrichment within the perivascular subarachnoid space of ACA (Fig. 5). Potentially, the arachnoid membranes in the subarachnoid spaces might as well be expected to create some barrier function; particularly, the Liliequist membrane could be hypothesized to serve as a barrier for tracer propagation

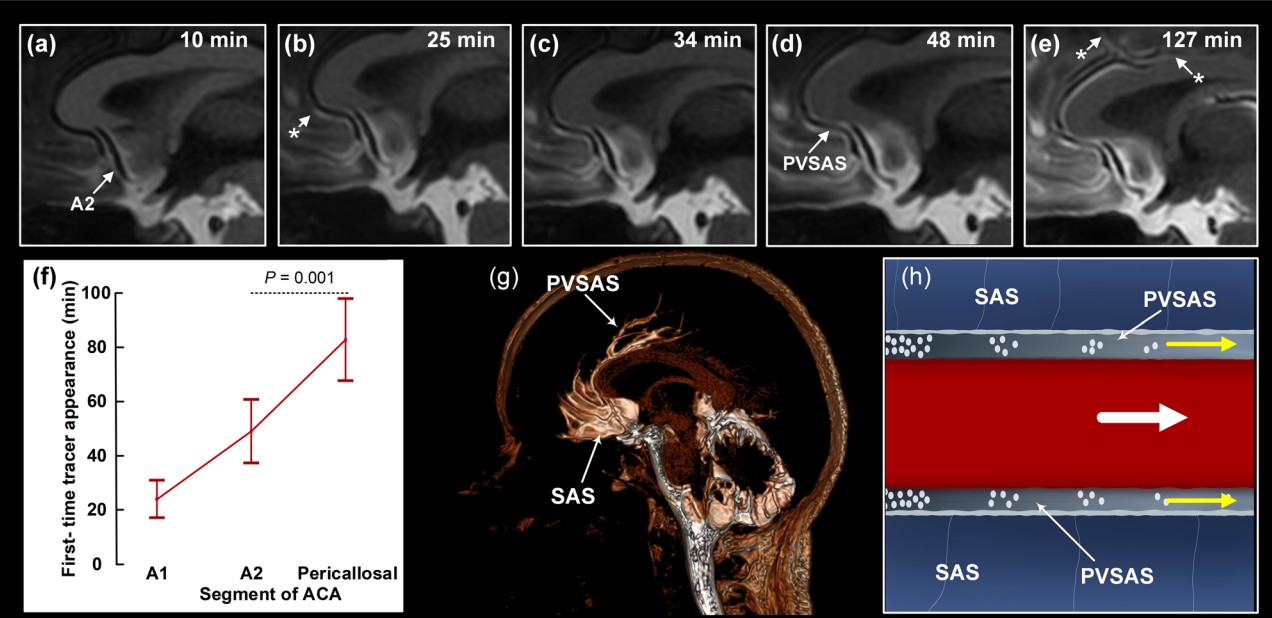

**Fig. 3 | Antegrade perivascular transport along the anterior cerebral artery (ACA) branches.** The time course of periarterial tracer enrichment in the perivascular subarachnoid space (PVSAS) along ACA (**a**–**e**) indicates antegrade tracer propagation. The asterisk indicates first-time appearance of tracer along ACA. **f** The time from intrathecal injection to first-time appearance of tracer along the different segments of ACA, i.e. A1 ($n = 11$), A2 ($n = 61$) and pericallosal artery ($n = 63$) is shown. Graph indicated by mean and 95% confidence intervals (CI) and differences determined by ANOVA with post-hoc Bonferroni corrections. Variation in spinal transit time was no confounder for differences in first-time appearance of tracer between the vascular segments. **g** A 3D image shows tracer enrichment in subarachnoid space (SAS), and along the perivascular subsrachnoid space (PVSAS) of the pericallosal artery (note the sparse distal enrichment as compared with the pronounced proximal enrichment of subarachnoid space). **h** A cartoon illustrates the tracer transport confined to the PVSAS, moving in antegrade direction (arrow); illustration: Øystein Horgmo, University of Oslo.

towards the perivascular compartment, though this was not apparent from the present images (Fig. 5). The propagation of tracer is further shown in Supplementary Movie 2. In Fig. 6 is shown a 3D visualization of tracer propagation, further supporting the view of free tracer propagation from thecal sac, via basal cisterns to perivascular subarachnoid space of supratentorial compartment. Further examples of tracer enrichment pattern within basal cisterns are given in Supplementary Figs. 7 and 8, and Supplementary Movie 3. In general, our present observations suggest unrestricted tracer passage within basal cisterns before entry into periarterial subarachnoid spaces.

There was no evidence of a compartmentalized subarachnoid space along the vertebral artery (VA) and basilar artery (BA) (Fig. 5; Supplementary Fig. 7, 8). Instead, the enrichment of the prepontine cistern occurred rapidly, with no sign of a perivascular VA/BA barrier.

### The periarterial subarachnoid transport precedes tracer enrichment in cerebral cortex

CSF enrichment of periarterial spaces within the cortex has been proposed as crucial for the metabolic clearance of solutes from brain[14]. To address whether tracer enrichment in the cerebral cortex is associated with first-time appearance of periarterial tracer enhancement in the subarachnoid space, we explored the association between tracer enrichment in the cerebral cortex after 2 h and first-time appearance of perivascular tracer enrichment along ACA and MCA, respectively. At group level, tracer enrichment in the cerebrum at 2 h occurred nearby the large artery trunks (Fig. 7a–c). Furthermore, faster first-time tracer appearance was accompanied with more pronounced tracer enrichment in brain parenchyma. Accordingly, there was a negative correlation between tracer enrichment in frontal cortex at two hours and first-time tracer appearance in the ACA segments A1 (Fig. 7d), A2 (Fig. 7e), and pericallosal artery (Fig. 7f). Likewise, there was a negative correlation between tracer enrichment in temporal cortex and first-time tracer appearance in the MCA segments M1 (Fig. 7g), M2 (Fig. 7h) and M3 (Fig. 7i). Furthermore, there were even stronger correlations between first-time appearance of periarterial tracer enhancement and gray matter sub-regions most adjacent to the artery trunks. Accordingly, tracer enrichment in caudal anterior cingulate cortex became reduced with late first-time appearance of tracer in the ACA branches A1 (Supplementary Fig. 9a), A2 (Supplementary Fig. 9b) or pericallosal artery (Supplementary Fig. 9c). Tracer enrichment in insula became reduced with delayed first-time appearance of tracer along the MCA branches M1 (Supplementary Fig. 9d), M2 (Supplementary Fig. 9e) or M3 (Supplementary Fig. 9f). Therefore, the parenchymal tracer enrichment is highly associated with the tracer enrichment in the periarterial subarachnoid space.

### The association between age and periarterial subarachnoid transport

We further examined whether molecular transport within periarterial subarachnoid spaces is associated with age. When considering the entire cohort of patients, with increasing age, the first-time appearance of tracer was delayed in perivascular subarachnoid spaces of the ACA segments A2 (Supplementary Fig. 10a) and pericallosal artery (Supplementary Fig. 10b), MCA branch M2 (Supplementary Fig. 10c), and M3 (Supplementary Fig. 10d). On the other hand, for the entire cohort, with increasing age, tracer enrichment became reduced in frontal cortex (Supplementary Fig. 10e) and temporal cortex (Supplementary Fig. 10f). However, the disease category was a confounder, and when considering the underlying disease, correlations became non-significant. A higher number of patients with larger variation in age is needed to conclude whether the pace of periarterial solute transport and accompanied enrichment within brain is age-related.

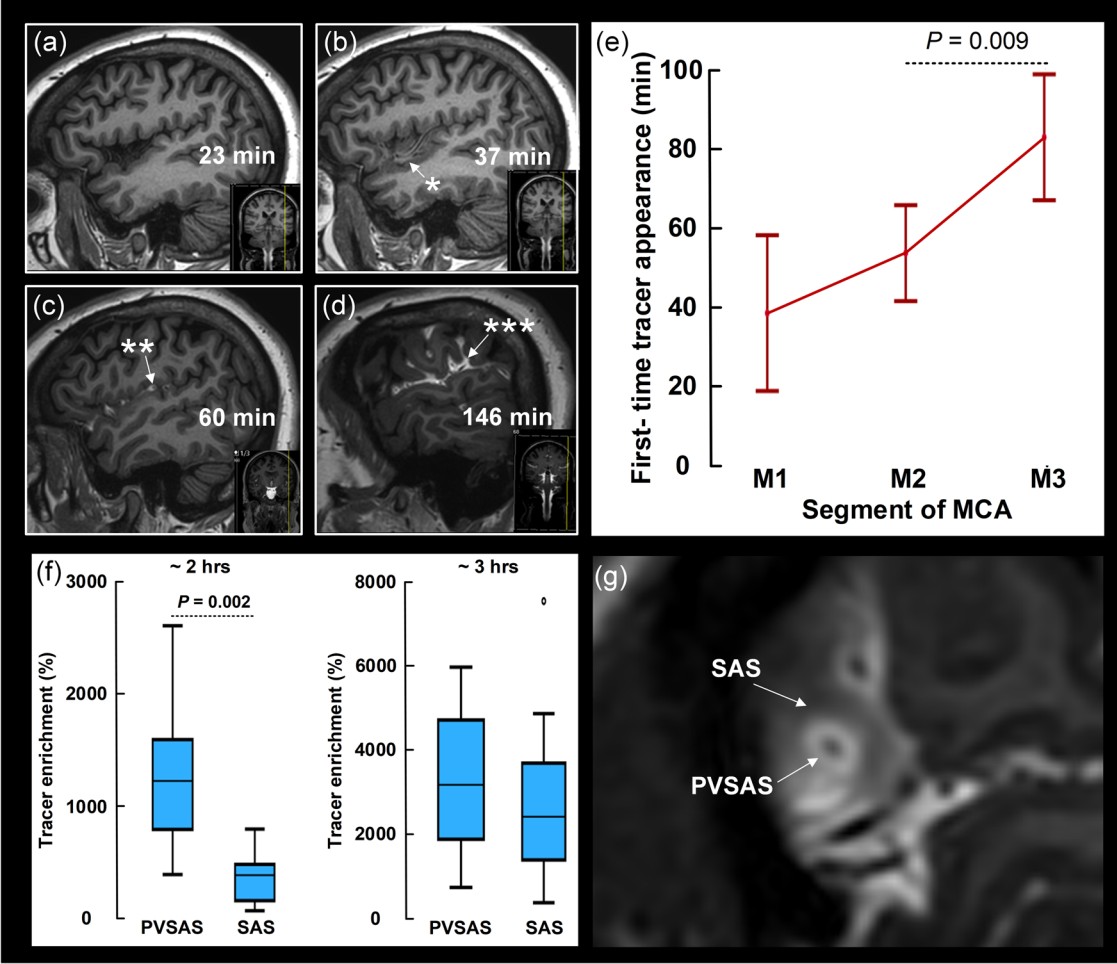

**Fig. 4 | Antegrade perivascular transport along the middle cerebral artery branches.** The time course of periarterial tracer enrichment in the perivascular subarachnoid space along MCA (**a–d**) indicates antegrade tracer propagation. The asterisk indicates first time appearance of tracer along MCA. **e** The time from intrathecal injection to first-time appearance of tracer along the segments of MCA, i.e. M1 ($n = 24$), M2 ($n = 68$) and M3 ($n = 59$) is shown. Graph indicated by mean and 95% confidence intervals (CI) and differences determined by ANOVA with post hoc Bonferroni corrections to correct for multiple comparisons. Variation in spinal transit time was no confounder for differences in first-time appearance of tracer between the vascular segments. **f** Measured at the M2 segment of the MCA about 2 h after injection, tracer enrichment was signficantly stronger in perivascular subarachnoid space (PVSAS; $n = 10$) than in surrounding subarachnoid space (SAS $n = 10$; $P = 0.002$; Mann-Whitney U-test). The difference was reduced at a later time (about 3 h; $n = 10$; $P = 0.63$; Mann-Whitney U-test). Box plots show median, 75% percentiles and ranges. **g** Visibly stronger tracer enrichment in PVSAS than surrounding SAS.

## The periarterial subarachnoid transport is impaired with reduced intracranial pressure-volume reserve capacity

We also asked whether tracer transport within periarterial subarachnoid spaces associate with the pulsatile ICP, which is a measure of the intracranial pressure-volume reserve capacity and a marker of the intracranial compliance[15]. Hence, the intracranial pulse pressure increases with reduced intracranial pressure-volume reserve (impaired compliance)[16]. In the clinical setting, we determine the mean ICP wave amplitude (MWA) from the continuous ICP measure during consecutive 6 s time windows (Fig. 8a) and computes the average of MWA during overnight ICP measurements. In this cohort, overnight average pulsatile ICP scores were available in 19 patients with the dementia subtype idiopathic normal pressure hydrocephalus (iNPH), four reference (REF) subject, seven with pineal cysts and two with communicating hydrocephalus. With increasing average MWA during overnight recording, indicative of reduced intracranial compliance, first-time appearance of tracer occurred later in the pericallosal ACA branch (Fig. 8b) and the M2 branch of the MCA (Fig. 8c). Furthermore, dichotomizing the overnight MWA scores into normal/abnormal categories according to previous criteria[17], demonstrated later first-time tracer appearance in pericallosal artery branch of ACA (Fig. 8d) and in M2 branch of MCA (Fig. 8e).

Accordingly, periarterial subarachnoid transport is impaired in patients with reduced intracranial pressure-volume reserve capacity.

## The perivascular subarachnoid tracer transport is impaired in a dementia subtype

Finally, we asked whether the periarterial subarachnoid space is functionally and anatomically affected by disease. Our cohort of 75 subjects included 14 subjects with no final diagnosis of CSF disease, referred to as REF subjects, and 22 iNPH cases. We found that periarterial subarachnoid spaces were enlarged in cases with this dementia subtype (Fig. 9a–e). It may be noted that the artery in iNPH cases seemed to be located exentric towards the border of the perivascular compartment (Fig. 9a–e). Morphological differences between REF and iNPH subjects are further shown in Supplementary Fig. 11. Comparison of REF and iNPH cases showed later first-time appearance of tracer in the ACA branches A2 and pericallosal artery (Fig. 9f). We also note that the time difference between the first appearance of tracer around pericallosal artery versus A2 branch of ACA was longer in iNPH than REF subjects [$61.3 \pm 39.5$ min ($n = 13$) vs. $13.0 \pm 13.7$ min ($n = 13$); $P = 0.009$; Mann-Whitney U-test]. Moreover, time of first periarterial tracer enhancement along the MCA branches M2 and M3 occurred

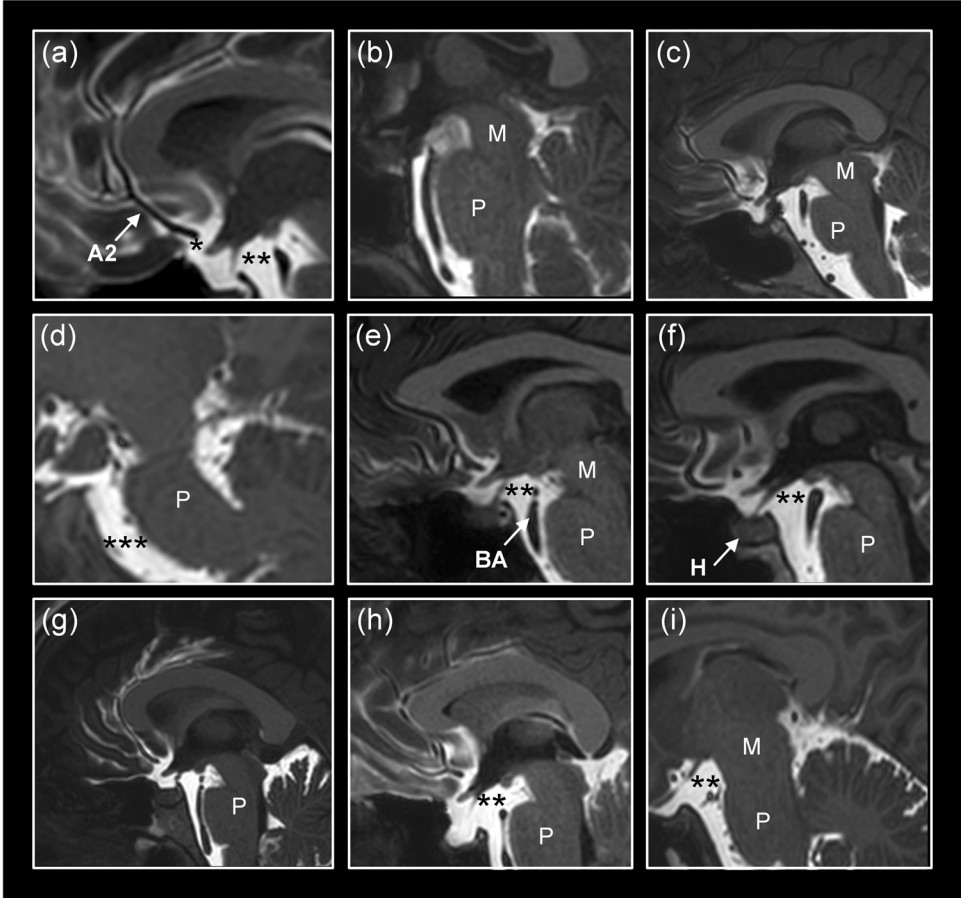

**Fig. 5 | Direct tracer propagation between subarachnoid basal cisterns and the perivascular subarachnoid space. a–i** The present observations gave evidence for direct communication between prepontine and interpeduncular cisterns and the perivascular subarachnoid spaces. Only one of 75 subjects demonstrated some barrier function of the Lilliequist membrane. Time from intrathecal tracer injection: **(a)** 132 min, **(b)** 34 min, **(c)** 15 min, **(d)** 14 min, **(e)** 22 min, **(f)** 30 min, **(g)** 9 min, **(h)** 140 min, and **(i)** 26 min. A2: A2 segment of anterior cerebral artery. BA: Basilar artery. H: Hypophysis. M: Mesencephalon. P: Pons. * Prechiasmatic cistern. ** Interpeduncular (or premesenchephalic) cistern. *** Prepontine cistern.

later in iNPH than REF subjects (Fig. 9g). Therefore, in subjects with the iNPH dementia subtype, enlarged periarterial subarachnoid spaces were accompanied with reduced pace of perivascular tracer transport. Furthermore, the area of perivascular subarachnoid space was significantly larger in iNPH cases (Fig. 9h). We also found that tracer enrichment in the cerebral cortex was lower in iNPH than REF subjects at 2 h, both for the frontal cortex (Fig. 9i) and temporal cortex (Fig. 9j).

In addition to the REF ($n = 14$) and iNPH ($n = 22$) subjects included in the study, the remaining 39 subjects had various types of other tentative CSF diseases (Supplementary Table 1). The occurrence of a perivascular tracer compartmentalization around the major arteries ACA, MCA, PCA and VA/BA is presented in Supplementary Table 2. Comparing first-time tracer appearance in the ACA branches A1, A2 and pericallosal artery, and the MCA branches M1, M2 and M3 disclosed some differences between groups (Supplementary Table 3; Supplementary Fig. 12). Furthermore, as expected from differences in periarterial tracer enrichment, the tracer enrichment in frontal and temporal cortex also differed significantly between groups (Supplementary Table 4; Supplementary Fig. 13). These observations suggest that underlying CSF disease may affect transport of substances in periarterial subarachnoid spaces and the subsequent entry of substances into brain parenchyma.

## Discussion

The present study provides in vivo evidence for the existence of a compartmentalized human subarachnoid space, constituted by a semipermeable perivascular membrane. Here, we primarily provide evidence for a periarterial subarachnoid space enabling transport of solutes in antegrade direction along the anterior-, middle-, and posterior cerebral arteries toward the cerebral cortex. The degree of tracer enrichment in cerebral cortex depends on the amount of periarterial tracer enrichment. Moreover, the periarterial molecular transport becomes impaired with reduced intracranial pressure-volume reserve capacity and may be impaired in diseases, here illustrated by the dementia subtype iNPH.

The observation that the tracer was concentrated circumferentially around the arteries, indicates a barrier, preceding diffuse enrichment in the surrounding subarachnoid space, suggesting the barrier is semi-permeable. We were unable to identify directly the membrane itself due to limitations in MRI resolution. To the best of our knowledge, there are no previous descriptions of how a compartmentalized, perivascular subarachnoid space affects CSF flow in humans. Previous studies on the arachnoid mater have primarily addressed the anatomical organization of arachnoid trabeculations and membranes[9,18]. For decades, the arachnoid anatomy has attracted the interest of neurosurgeons; Yassargil[19] introduced the technique of moving micro-surgically from one arachnoid cistern to another, using these as surgical landmarks. Rhoton studied the organization of arachnoid cisterns such as the prepontine, interpeduncular, and ambient cisterns, and how blood vessels and cranial nerves traverse them[20]. Most attention has been given to the anatomy of the Lilliequist membrane, which was first described by Key and Retzius in 1875[21], and

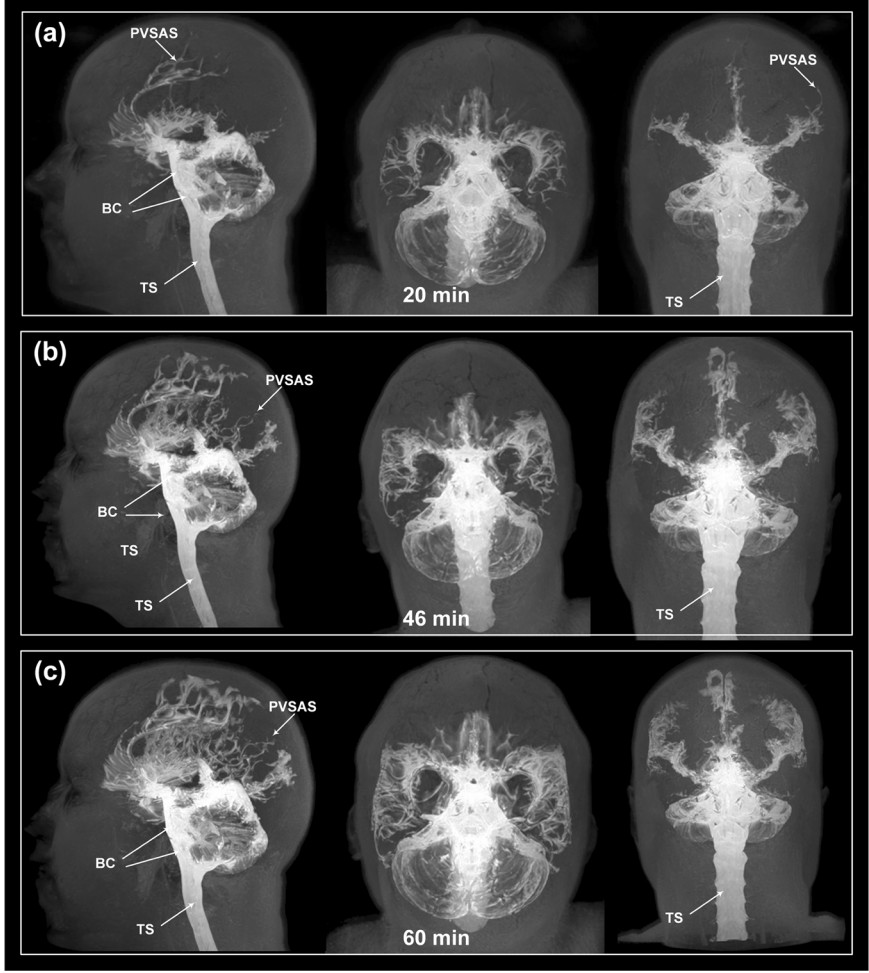

**Fig. 6 | 3D representations of direct tracer propagation between subarachnoid basal cisterns and the perivascular subarachnoid space. a–c** The present observations suggest direct passage of tracer from the thecal sac (TS), via basal cisterns (BC) towards the perivascular subarachnoid spaces (PVSAS). The 3D images show tracer enrichment in sagittal, axial and cornal planes, assessed 20 min (**a**), 46 min (**b**) and 60 min (**c**) after intratehcal tracer injection.

rediscovered by the Swedish neuroradiologist Liliequist in 1956[22], which has later been shown to be created by mesencephalic and diencephalic leaves[23]. The anatomy of this membrane has been thought to impact CSF flow and is considered by neurosurgeons performing endoscopic third ventriculostomy for non-communicating hydrocephalus[23]. It is worth noting that the current CSF tracer freely passed the Liliequist membrane, with no evidence of compartmentalization of the tracer within the prepontine or interpeduncular cisterns. Rather, the tracer seemed to pass without restriction towards the perivascular spaces of the arterial trunks of the ACA and MCA via the suprasellar cistern and to the PCA via the ambient cistern. We cannot exclude the existence of a perivascular arachnoid barrier surrounding the vertebral and basilar arteries, though we were not able to demonstrate it.

At the ultrastructural level, the leptomeninges, consisting of the arachnoid membranes and the pia mater, were previously shown to cover arteries within the subarachnoid space[24,25], though their functional implications have been unknown. More recently, CSF tracer studies exploring the glymphatic system examined perivascular tracer transport in subpial arteries[14,26], but not in larger arteries residing within the subarachnoid space. The periarterial membrane referred to here may possibly compare with the pial sheath referred to by Zhang et al.[25]. These authors reported that arteries, not veins, in the subarachnoid space were coated by a thin sheath of outer pia mater cells, creating a periarterial space. They further speculated whether the

periarterial space might function as a drainage pathway for interstitial fluid from brain tissue, i.e. allowing retrograde transport in a proximal direction along arteries residing within the subarachnoid space. This assumption is, however, not supported by the present observations, which rather suggest antegrade transport of tracer along arteries towards the brain.

Concerning the periarterial versus perivenous subarachnoid space, in this present study, enhancement around a larger vein occurred in only few subjects, and typically after the diffuse enhancement pattern in subarachnoid space had already appeared. In our view, the reason why we did not regularly observe perivenous enhancements is not due to methodological limitations, but rather evidence that periarterial propagation of CSF tracer in subarachnoid space is far most important, at least dominating over propagation along veins.

The present observations are somewhat difficult to interpret with regard to the recently described fourth meningeal layer, denoted SLYM[8]. This layer was proposed to segregate the subarachnoid space into two, an outer and inner layer, justifying the claim of a fourth membrane. Comparably, this discovery also heavily relied on tracer administration in CSF and observations of subsequent accumulation of CSF tracer around subpial arteries. In the present human study, we made observations of a periarterial compartment within the subarachnoid space, where a dichotomization into an outer and inner layer seems less meaningful, since arteries may also be located within the inner parts of subarachnoid space towards the pia mater. Whether

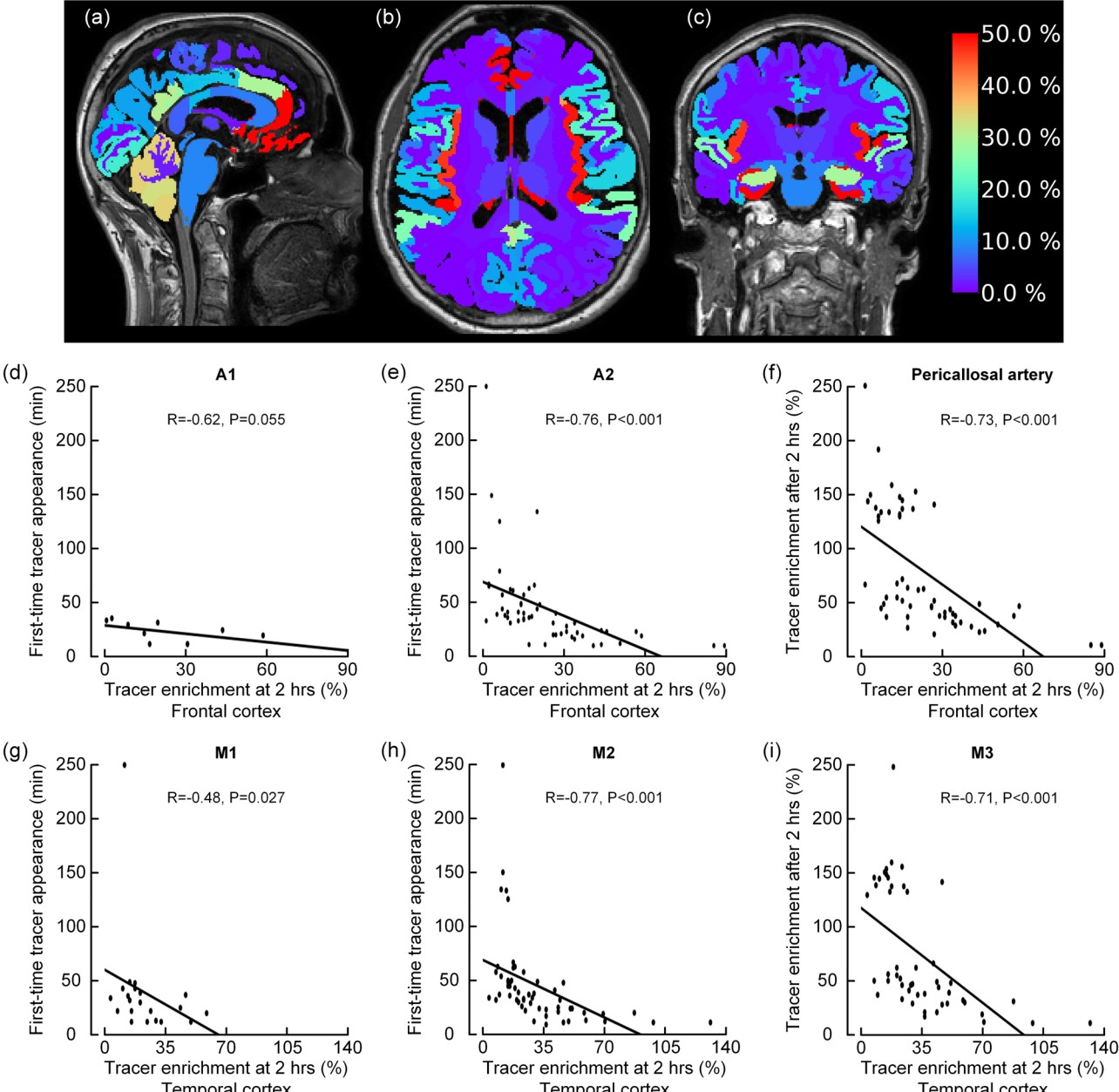

**Fig. 7 | Periarterial subarachnoid tracer transport precedes tracer enrichment in cerebral cortex.** After being confined to the periarterial space, the tracer passed to the surrounding subarachnoid space (SAS) and further to the extravascular compartment within the cerebral cortex. **a–c** Sagittal, axial, and coronal MRI shows tracer enrichment in brain as percentage change in normalized T1 signal at 2 h after intrathecal tracer (gadobutrol, 0.50 mmol) administration (percentages shown on the color bar to the right). Tracer enrichment in CSF is removed to show tracer enrichment in brain only. Highest tracer enrichment (red color) is seen cerebral cortex nearby the ACA (**a, b**), MCA (**b, c**) and PCA (**c**). **d–f** The correlation between tracer enrichment in frontal cortex (gray matter) and first-time appearance of tracer in A1, A2 and pericallosal artery segments of ACA. **g–i** The correlation between tracer enrichment in temporal cortex (gray matter) and first-time appearance of periarterial tracer enhancement in M1, M2 and M3 segments of MCA. The negative correlations show that shorter first time appearance of tracer was associated with stronger tracer enrichment in cerebral cortex. For the individual plots, the Spearman correlation coefficient is given with significance level, and fit line shown. Images (**a–c**): Lars Magnus Valnes, Department of neurosurgery, Oslo University Hospital-Rikshospitalet.

our observations add to the previous report of SLYM[8], which tentatively might overlap with, or correspond to, the outer pial layer, should be subject to future studies. Our study concludes about the existence of a perivascular subarachnoid space, abbreviated PVSAS, surrounding larger arteries at the surface of the gyrencephalic brain, delineated by a semipermeable membrane that aligns well with the barrier threshold of 3 kDa described for SLYM, which also was impermeable to 1 μm wide fluorescent particles.

Concerning the barrier function of the currently described periarterial compartment, its membrane seems semipermeable to the 604 Da molecular weight tracer since the tracer was contained within the perivascular space only for a limited time. The previously described pial sheath covering arteries within the subarachnoid space was as well proposed to be permeable to solutes, though functional data were not given[25]. Other researchers reported that leptomeningeal sheaths surrounding blood vessels in the subarachnoid space contain stomata (<10–20 μm) with underlying perivascular connective tissue[27]. However, to which degree 10–20 μm sized stomata would functionally affect a periarterial membrane barrier, is an open question. It should be noted that the currently used tracer is a

hydrophilic substance lacking transmembrane transporter molecules and is mainly contained outside the intact BBB when administered intrathecally, allowing it to be restricted from leakage into systemic blood circulation. Our previous findings reported a cranially directed distribution of the tracer after intrathecal lumbar injection, with the tracer entering the cisterna magna after an average of $20 \pm 23$ min[28]. For the present cohort, the spinal transit time was even shorter, at $13.4 \pm 6.3$ minutes.

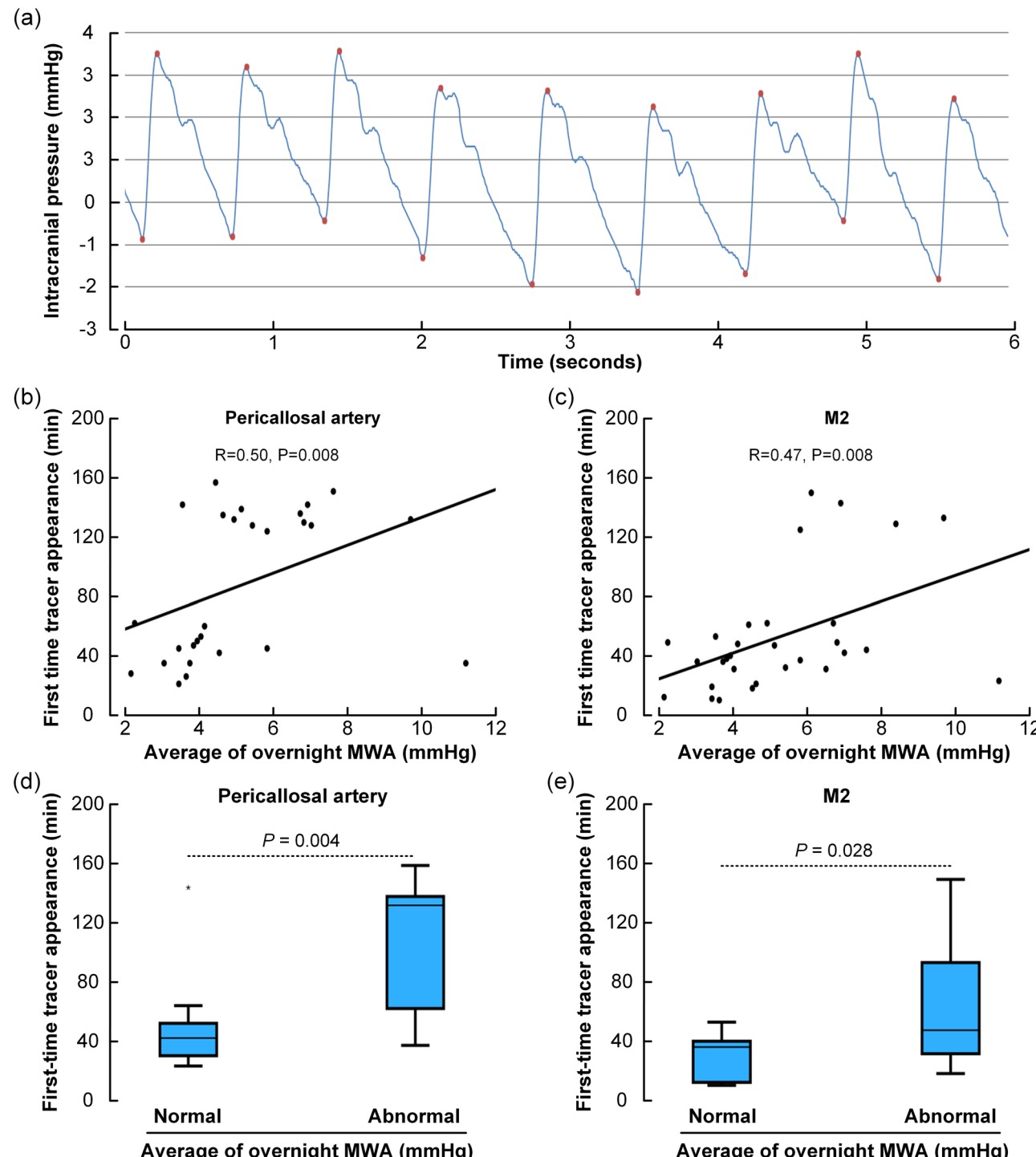

**Fig. 8 | The periarterial subarachnoid tracer transport depends on the pulsatile intracranial pressure (ICP). a** The pulsatile intracranial pressure (ICP) refers to the cardiac-induced pressure waves from the continuous ICP signal, which was quantified as the mean ICP wave amplitude (MWA) over consecutive 6-seconds time windows. The average of MWA during overnight ICP measurements was calculated. With increasing pulsatile ICP, the perivascular tracer transport became slowed down. Thus, there was a significant positive correlation between average of overnight MWA and first-time tracer appearance in (**b**) pericallosal artery of anterior cerebral artery (ACA), as well as (**c**) M2 segment of middle cerebral artery (MCA). For the individual plots, the Spearman correlation coefficient is given with significance level, and fit line shown. Variation in spinal transit time was no confounder for correlations in **b** and **c**. Furthermore, dichotomizing over-night MWA scores as Normal ($n = 10$) and Abnormal ($n = 22$) according to previously described criteria[17] showed in subjects with abnormal elevated MWA significantly delayed first-time tracer appearance in pericallosal artery segment of ACA (**d**) and M2 segment of MCA (**e**). Box plots show median, 75% percentiles and ranges. Statistical differences determined by Mann-Whitney U-test.

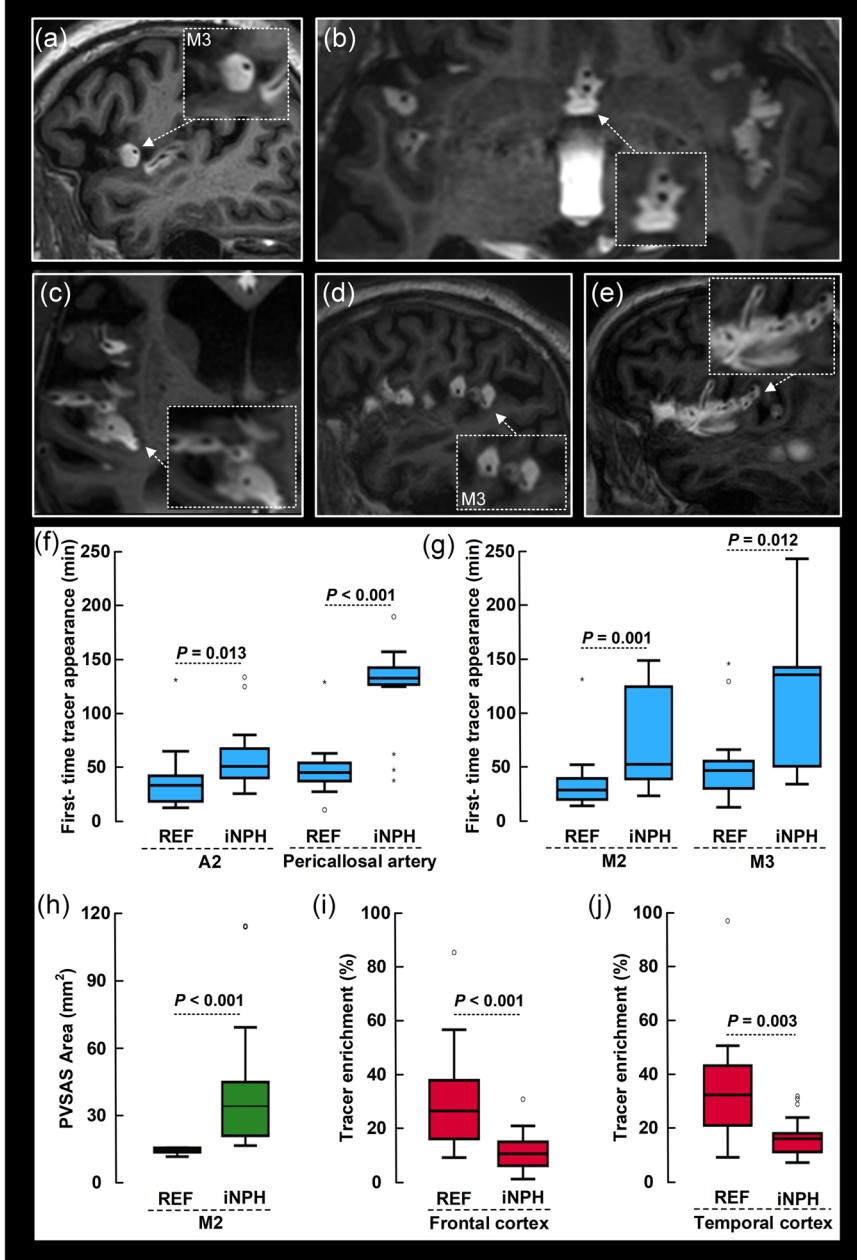

**Fig. 9 | Alterations of perivascular subarachnoid spaces in subjects with a dementia disease.** The dementia subtype idiopathic normal pressure hydrocephalus (iNPH) presents with enlarged and more irregular perivascular subarachnoid spaces (PVSAS) **a–e**. Time from intrathecal tracer injection: **a** 54 min, (**b**) 33 min, (**c**) 203 min, (**d**) 130 min, and (**e**) 130 min. The morphological alterations of PVSAS were accompanied with delayed perivascular tracer transport. First-time appearance of tracer occurred later in the ACA branches A2 of iNPH ($n = 15$) than reference (REF) subjects ($n = 13$), in pericallosal artery of iNPH ($n = 16$) than REF ($n = 13$) subjects, in pericallosal artery of iNPH ($n = 16$) than REF ($n = 13$) subjects (**f**), as well as in the MCA branches M2 of iNPH ($n = 18$) than REF ($n = 13$) subjects and in M3 of iNPH ($n = 13$) than REF ($n = 13$) subjects (**g**). The area of PVSAS in the M2 segment was larger in iNPH patients ($n = 19$) as compared with REF ($n = 9$) individuals (**h**). Furthermore, tracer enrichment in gray matter at 2 h was reduced in iNPH ($n = 22$) than REF ($n = 14$) in frontal cortex (**i**) and temproal cortex (**j**). Box plots show median, 75% percentiles and ranges. Signifcant differences between groups determined by Mann-Whitney U-test.

Here, we conclude that periarterial tracer transport occurred in the antegrade direction along the branches of the ACA, MCA, and PCA. We provide several lines of evidence for this statement: i) Tracer appeared significantly later in pericallosal than A2 segments of ACA (Fig. 3f). ii) Tracer appeared significantly later in M3 than M2 segments of MCA (Fig. 4e). iii) At the M2 segment of the MCA after 2 h, the tracer enrichment was significantly stronger in the perivascular subarachnoid space than the surrounding subarachnoid space, suggesting that tracer transport is facilitated along arteries. iv) Shorter first-time appearance of tracer along the ACA and MCA correlated significantly with tracer

enrichment in respective cortical areas (Fig. 7d–i; Suppl Fig. 9a–f). These observations suggest that flow in the subarachnoid space is directional, facilitated by a perivascular barrier, enabling transport of solutes along the arterial tree towards the brain. Several previous rodent studies utilizing tracers and particles have provided evidence for antegrade perivascular transport of substances around subpial arteries at the brain surface[26,29]. On the other hand, antegrade perivascular molecular transport at larger scale along arteries within the subarachnoid space was previously not reported. While the study setup does not exclude CSF flow in the retrograde (peripheral to proximal) direction, an intrathecally

injected tracer will in the cisterna magna blend with ventricular CSF and follow the physiological flow patterns within the intracranial compartment. The presently described procedure of intrathecal injection of 0.5 mmol gadobutrol mixed with saline in a total volume of 1 ml would not be expected to cause changes in ICP or CSF flow.

Periarterial CSF flow is proposed to be a crucial entry point for the glymphatic circulation of the brain[14]. The presently described compartmentalization may thus enable faster and more efficient transport of substances to the brain than from the surrounding subarachnoid space, where solute transport may be less directional. In line with this, there was a significant association between the degree of perivascular tracer enrichment and tracer enrichment within the cerebral cortex. Hence, the reduced amount of perivascular tracer was accompanied with reduced entrance of tracer to the cerebral cortex. In this regard, the present observations that the perivascular tracer transport is highly individual is of utmost interest. The glymphatic pathway may be crucial for intrathecal drug delivery to the central nervous system (CNS). The intrathecal route is increasingly used to treat neurological diseases[30] and enables for small body doses of potentially toxic drugs while allowing for by-passing the blood-brain-barrier by penetration into brain tissue directly from the surface. The function of the periarterial transport may in this regard be crucial for the efficacy of CSF-mediated drug delivery to CNS. Møllgård et al.[8]. reported that damage to SLYM impairs periarterial solute transport. Others have previously shown that aging was accompanied with inflammation and fibrosis of the arachnoid membrane[31]. These observations raise the question of what happens in disease, e.g. after events such as subarachnoid hemorrhage (SAH) and traumatic brain injury (TBI), which are known to impair glymphatic transport[32]. In the present study, for the total cohort, perivascular tracer transport became slower with increasing age, but results were confounded by diagnosis. We also showed reduced tracer in the cerebral cortex at 2 h with increasing age, independent of diagnosis, which compares with previous experimental rodent data about impaired glymphatic function with increasing age[33].

An important observation was that perivascular molecular transport became impaired with reduced intracranial pressure-volume reserve capacity. From before, it has been shown that arterial hypertension, by its infliction on reduced arterial wall pulsatility, causes reduced periarterial tracer movement[26]. It should be noted that intracranial and vascular pressure pulsatility are normally not correlated. In previous studies, we found that high correlation between intracranial and vascular pressure pulsatility in patients with subarachnoid hemorrhage was accompanied with worse outcome[34]. In subjects with iNPH, there was low correlation between intracranial and vascular pressure pulsatility in the majority of patients[35]. We suggest that elevated intracranial pulsatility due to underlying disease causes restriction on the arterial pulsatility, which in turn hampers the perivascular tracer transport. No studies have previously addressed the role of the milieu surrounding the perivascular spaces in larger context. Our observations suggest that the biophysical milieu of the brain, here assessed by the pulsatile ICP, reflecting the pressure-volume reserve capacity, also affects perivascular molecular transport. With increasing pulsatile ICP (i.e. increasing MWA), intracranial compliance is reduced[16], which in turn slows down the perivascular molecular transport. Perhaps impaired intracranial compliance restricts the arterial pumping as in arterial hypertension, which in turn hampers the driving forces of perivascular molecular transport.

Another intriguing observation of this study was the morphological alterations of periarterial subarachnoid spaces in the dementia subtype iNPH. These patients showed wider periarterial spaces, which were combined with a reduced pace of perivascular molecular transport and reduced enrichment of the cerebral cortex. The dementia subtype iNPH was previously shown to have impaired glymphatic tracer clearance[11]. The present findings of differences in first-time appearance of periarterial tracer between patients with various CSF

diseases further point to the association between underlying disease and periarterial subarachnoid space transport.

We acknowledge some limitations of this study. The functional evidence presented was based on the first-time appearance of the tracer within the perivascular space, and the time resolution of the MRI acquisitions was limited. Events occurring between MRI scans could therefore have been missed, thus we cannot be sure whether enrichment of the outer subarachnoid space compartment occurred directly from the basal cisterns or from tracer that leaked through a semipermeable perivascular barrier. Furthermore, spinal transit time for the intrathecal tracer varied, rendering for different degrees of intracranial enrichment at same time points after injection, which may have affected our ability to detect tracer enhancement along vessels more distally. However, we only included patients with spinal transit time less than 35 min to mitigate this variation, and with the presently used time resolution, we were able to answer the research questions. Additionally, the administration of gadobutrol intrathecally is currently off-label and was used exclusively in patients on clinical indication. The ability to perform repeated MRI scanning in healthy subjects is therefore limited. Nevertheless, 14/75 patients were not diagnosed with any CSF disorder after an extensive clinical- and MRI work-up and can thus be considered close to healthy. It may as well be considered a limitation that the assessment of MRI was not blinded. While blinding would be preferable it might hardly be achievable since images are rather typical for the diagnoses, e.g. ventriculomegaly in iNPH and communicating hydrocephalus. On the other hand, the neuroradiologist (GR) assessing images did not consider patient diagnosis and assured that measurements were performed in a standardized manner in all patients to limit biasing.

Furthermore, we are unable to draw conclusions about the cellular properties of a perivascular arachnoid barrier, nor whether this barrier adds to the previously described SLYM[8], which was proposed to differ from other arachnoid cell layers due to its cellular composition and immunological profile. The authors of that study[8] found that the SLYM contains immune cells and markers of lymphatic endothelial cells (LYVE-1, PROX-1, and PDPN) lining the SLYM, with a high content of myeloid CD45+ cells (macrophages) that increases with age. An intriguing question is the possible role of the periarterial subarachnoid barrier as an immune interface. Generally, mesothelial membranes act as immune barriers[36]. Future studies need to address whether neuroinflammatory conditions might alter the content of immune cells in the subarachnoid space, as these cells produce cytokines and reactive oxygen substances that could damage the periarterial subarachnoid barrier and thereby affect periarterial solute transport. To this end, it was recently shown that sphingosine-1-phosphate, which is contained within the arachnoid membrane regulated MCA vasoconstriction[37], suggesting a role of the arachnoid membrane in regulation of cerebral vascular tone. Further studies are required to determine the cellular composition of the human perivascular subarachnoid barrier reported in the present study.

In conclusion, the present study provides evidence for the existence of compartmentalized CSF flow within the subarachnoid space delineated by a semipermeable membrane surrounding artery trunks outside the gyrencephalic human brain. This perivascular subarachnoid space allows for directed, antegrade transport of the tracer along arteries, which precedes, and is associated with, tracer enrichment in the adjacent cerebral cortex. The periarterial molecular transport within subarachnoid space is impaired with reduced intracranial pressure-volume reserve capacity and in the dementia subtype iNPH. Further investigations are necessary to determine the cellular constituents of this perivascular subarachnoid membrane and how it is affected by disease.

## Methods
### Approvals
The following authorities approved the study: The Institutional Review Board (2015/1868), Regional Ethics Committee (2015/96) and the

National Medicines Agency (15/04932-7), and registered in Oslo University Hospital Research Registry (ePhorte 2015/1868). Inclusion was by written and oral informed consent from all participants. They received no compensation.

## Experimental design

The study followed a prospective and observational study protocol. Randomization of patients or a priori sample size calculation are not relevant.

## Patients

The study was performed in individuals consecutively included after referral to the Department of neurosurgery, Oslo University Hospital - Rikshospitalet, Oslo, Norway, for work-up of tentative CSF circulation disorders (Supplementary Table 1). The presently included cohort consisted of 75 subjects (28 males and 47 females) in whom a series of repeated MRI acquisitions were done during the first two hours after intrathecal contrast administration. Patients with a spinal transit time of CSF tracer >35 min were not included in the analysis; spinal transit time refers to the time between intrathecal tracer injection and the appearance of tracer in cisterna magna.

## MRI protocol

All MRI scans were obtained using a 3 Tesla MRI unit (Philips Ingenia®) with a 32-channel head coil; equal imaging protocol settings were used at all time points when sagittal 3D T1-weighted volume scans were acquired. MRI acquisitions were done continually with the patient in the scanner the first hour, and then repeated at 2–3 h after i.th. injection. Patients were supine during the entire period. The patient stayed in the MRI lab, and during MRI scanning there was regular contact between patient and MRI technician. Between each scan, lasting about 5 min, there was always a short oral communication between the patient and the MRI technician running the exam. No patients went to sleep during stay in the MRI lab.

The following imaging parameters were applied: Repetition time = "shortest" (typically 5.1 ms), echo time = "shortest" (typically 2.3 ms), Flip angle = 8 degrees, field of view = 256 × 256 cm and matrix = 256 × 256 pixels (reconstructed 512 × 512). Hundred and eighty-four over-contiguous (overlapping) slices with 1 mm thickness were obtained that were automatically reconstructed to 368 slices with 0.5 mm thickness. The imaging stacks for each time point were planned to use an automated anatomy recognition protocol based on landmark detection in MRI data (SmartExam™, Philips Medical Systems, Best, The Netherlands) to secure consistency and reproducibility of the MRI slice placement and orientation.

The MRI acquisitions were repeated continuously during the first hour and then repeated two hours after intrathecal injection of 0.5 ml gadobutrol (1 mmol/ml) (Gadovist®, Bayer AB, Sweden), which was performed by an experienced interventional neuroradiologist under fluoroscopic guidance at the lower lumbar level. The correct needle (22 G) position within the lumbar subarachnoid space was confirmed by backflow of CSF. Patients were in the supine position when the MRI acquisitions were done.

All MRI scans were visually assessed in a standardized manner by a board-certified radiologist with 16 years' experience in neuroradiology (G.R.). Assessment was not blinded. MRI scans were read directly from the hospital picture archiving and communication system (PACS, SECTRA®, Sweden).

For assessment of first-time appearance of tracer along the arterial branches, CSF tracer-enhanced T1-weighted volume scans were scrutinized in three orthogonal imaging planes for signs of a compartmentalized subarachnoid space within the basal cisterns and along the major artery trunks traversing the brain surface, where patterns of tracer distribution were noted based on visual assessment. Time point of first tracer enhancement in a perivascular fashion was

registered for different segments of the anterior cerebral artery (A1, A2, pericallosal artery) and middle cerebral artery (M1, M2, M3) in image slices orthogonal to the vessel orientation. Any presence of a similar pattern of tracer enhancement along the posterior cerebral artery was also noted. Examples of regions of interest (ROIs) in peri-vascular subarachnoid space and subarachnoid space over time are shown in Supplementary Fig. 14.

We compared area of perivascular subarachnoid space between REF and iNPH subjects using a method shown in Supplementary Fig. 15.

For estimation of contrast enrichment in the frontal and temporal cortex, segmentation was performed using the FreeSurfer software (version 6.0) (http://surfer.nmr.mgh.harvard.edu/) for segmentation, parcellation and registration/alignment of the longitudinal data to investigate the increase of T1 intensity due to CSF tracer[11]. Non-brain tissue was removed using a hybrid watershed/surface deformation procedure[38], automated Talairach transformation, segmentation of the subcortical white matter and deep gray matter volumetric structures[39,40]. The median T1 signal unit for each time point was computed for each segmented area. Moreover, the median signal unit was divided with the signal unit of a reference ROI placed within the posterior part of the orbit in axially reconstructed images from the same T1 volume scan, the ratio referring to *normalized T1 signal unit*s, correcting for any baseline changes of image grey scale due to image scaling.

## Pulsatile intracranial pressure

Continuous overnight monitoring of pulsatile ICP was performed using a solid ICP sensor placed in the parenchyma (Codman ICP microsensor, Codman, Johnson & Johnson, Raynham, MA, USA) via a small burr hole, with online measurements of the pulsatile ICP (mean ICP wave amplitude; MWA) utilizing Sensometrics software (dPCom, Oslo, Norway). The Sensometrics software was not developed for this particular project, but has been utilized by Oslo University Hospital for more than ten years for on-line analysis of continuous ICP measurements (contact@dpcom.com). Measurements were not accompanied with any CSF drainage or any placement of catheter to the CSF space. The MWA is computed over 6-seconds time windows and refers to the pressure changes occurring during the cardiac cycle (Fig. 8a). The average values of MWA were computed during overnight monitoring; abnormal MWA refers to average of overnight MWA > 4 mmHg and >5 mmHg in >10% of recording time[16,17].

## Statistical analysis

Statistical analyses were performed using the SPSS software version 29 (IBM Corporation, Armonk, NY). Differences between groups were tested with Mann-Whitney U-test when no normal distribution of data was present. Analysis of variance (ANOVA) with Bonferroni-corrected post-hoc test was used for multiple comparisons. Correlations were determined by Spearman's correlation coefficient. Differences between categorical data were determined by Pearson Chi-square test. When appropriate, we tested possible confounder effects on observed correlations between variables. Statistical significance was accepted at the .05 level; we exclusively used two-sided tests for all statistical analyses.

## Reporting summary

Further information on research design is available in the Nature Portfolio Reporting Summary linked to this article.

# Data availability

All data supporting the findings of this study are available within the paper and in supplementary information files. The MRI measurements for all included ROIs of the study are available in the Source Data file. Raw data of the MRI measurements are not publicly available according to privacy guidelines of Oslo University Hospital. Anonymized images and any additional raw data are available from the

corresponding author (Eide, Per Kristian, p.k.eide@medisin.uio.no) upon request, which conform with the privacy guidelines of the Oslo University Hospital. The timeframe for response to such requests is within 1-3 months. Source data are provided with this paper.

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

## Acknowledgements

This work was supported by grants from Health South-East, Norway (grants 2020068), and from Department of neurosurgery, Oslo university hospital-Rikshospitalet, Oslo, Norway. The authors thank dr. Øivind Gjertsen, dr. Bård Nedregaard and dr. Ruth Sletteberg from the

Department of Radiology, Oslo University Hospital – Rikshospitalet, who performed the intrathecal gadobutrol injections in all study subjects. We also sincerely thank the Intervention Centre and Department of neurosurgery at Oslo University Hospital Rikshospitalet for providing valuable support with MR scanning and care-taking of all study subjects throughout the study. The author thanks Lars Magnus Valnes, PhD, Department of Neurosurgery, Oslo University Hospital, Oslo, for Free-Surfer analysis of tracer enrichment of brain tissue, and for providing Supplementary Movie 2. We thank Are Hugo Pripp, Ph.D, Department of Biostatistics, Epidemiology and Health Economics, Oslo University Hospital, Oslo, for statistical advice. We thank Tomas Sakinis, M.D., for assistance with Supplementary Movie 1 and 3.

## Author contributions

Conceptualization and experimental design: PKE, GR. Imaging analysis: GR. Data analysis: PKE. Writing-original draft: PKE. Both authors contributed to review and revisions.

## Competing interests

PKE has a financial interest in the software company (dPCom AS, Oslo) manufacturing the software (Sensometrics Software) used for analysis of the ICP recordings. The other author (GR) declares no competing interests.
