## [Peer Review File · Nature Communications]

REVIEWER COMMENTS

Reviewer #1 (Remarks to the Author):

This is an impressive study that presents a large volume of CSF-contrast-agent timeseries MRI data in order to better understand CSF dynamics in the human brain. The methods required to generate these type of data are technically v. challenging, and the large volume of high quality data that has been generated speaks to the world-leading specialist knowledge and experience that the authors have in these methods (ie lumbar delivery of CSF contrast agents in patients combined with dynamic MRI).

The manuscript presents new and exciting data that convincingly demonstrates novel characteristics of CSF physiology in the human brain. Specifically, to my knowledge, this data provides new evidence in support of para-arterial CSF transport in the human brain and that this is related to the degree of CSF-contrast agent delivery to the brain tissue. Thus, it appears that the exciting conclusion reported in the title of the paper is well supported by the novel data presented here.

The manuscript is well written and the authors appears to have given a good amount of care to the interpretation of a large a highly complex data set.

There are however some important issues that I believe should be addressed:

- Figure 7. I think the ageing correlation is currently problematic as across the ~75 patients there are a number of different CSF-related conditions which have been diagnosed (table 1). These conditions are likely to be related to the age of the patient and so could be confounding the association with age that the authors seek to investigate. I recommend the authors consult a statistician to find the appropriate statistical method to test the effect of age on these MRI-based metrics given the underlying ~7 different CSF-related disease categories that exist within the data. The plots should also be color coded according to disease category to better communicate this important aspect of the data to the reader. Could the authors just look at correlation with age across the 14 ref subjects? I may have missed this but was spinal transit time accounted for in these measurements?

- 'All MRI scans were visually assessed in a standardized manner by a board-certified radiologist 424 with 16 years' experience in neuroradiology (G.R.)' – It was not clear from the description of the methods whether the analysis was blinded or not. This is important especially given the subjective nature of several of the analysis approaches where blinding should be employed.

-Results: 'Furthermore, diffuse tracer enhancement within the subarachnoid space was typically preceded by a rim of tracer enrichment surrounding the arteries residing within the subarachnoid space... The early dissipation of the periarterial enhancement pattern'. It would help to generate some more quantitative data to support this qualitative observation. Could the authors draw ROIs in the PVS region and subarachnoid region and show that the first appearance of the tracer is indeed sooner in the PVS region vs subarachnoid space?

- Results: 'As shown in Supplementary Figure 4, a perivenous pattern of tracer enhancement was occasionally detected, but veins at the brain surface are generally harder to depict at MRI, and thus also perivenous enhancement. However, the dataset is highly suggestive that the foremost front of perivascular tracer enhancement in subarachnoid space is primarily periarterial, not perivenous' – from these two sentences, the reader is left unclear whether the lack of evidence for perivenous enhancement is reflective of methodological limitations (ie veins are harder to depict with MRI) or genuinely reflects the underlying physiology. Can the authors explain/substantiate their summations in more detail as this is an important/interesting finding.

- Where possible, I think it would really help communicate the main findings of the work (eg. rapid Paraarterial CSF transport) by including videos showing the dynamic propagation of the tracer in the CSF spaces and ideally how this relates to the delivery of the tracer to the tissue.

- 'directional periarterial transport' – can the authors clarify/substantiate why they think that their data shows that periarterial transport is directional and their measurements do not reflect rapid pulsatile-driven mixing/dispersion (with no net directionality) driving the contrast agent down a concentration gradient? (ie what is the evidence for antegrade transport)?

- The intro is generally clear and well written. There have been a number of studies using gadolinium tracers in the CSF with dynamic T1 weighted MRI (many of which have been led by the authors of the current work), in the intro could the authors state more clearly/specifically how the design of the acquisition/ analysis in this work allows them to go above and beyond what is in the literature to address the aims of their work? ie what new question(s) does the design of the experiment allow to be addressed and why? (e.g more timepoints or higher spatial resolution of the images etc).

- For the 'intracranial pressure-volume reserve capacity' and 'in a dementia subtype' investigation of the MRI metrics (e.g 'first appearance'), I may have missed this but can the authors confirm/clarify that the spinal transit time was accounted for in their analysis?

- Figure 3. The analysis underlying the data presented in this figure is not described in the methods. Can the authors please add this and also clarify what question/hypothesis they were seeking to address with their analysis/ statistical tests employed here?

- Figure 6. For the estimates of parenchymal enhancement, it is important to rule out partial volume of the CSF space within the parenchymal ROIS – did the authors consider this in their analysis?

- Can the authors generate quantitative data or show more images to support their observation of increased PVS volume in INPH patients. Currently there just appears to be one representative image in support of this observation (supp Figure 9)?

- Methods: 'The MRI acquisitions were repeated continuously during the first hour and then repeated two hours after intrathecal injection of 0.5 ml gadobutrol (1 mmol/ml) (Gadovist® 417 , Bayer AB, 418 Sweden), which was performed by an experienced interventional neuroradiologist under fluoroscopic guidance at the lower lumbar level. Correct needle (22G) position within the lumbar subarachnoid space was confirmed by backflow of CSF. Patients were in the supine position when the MRI acquisitions were done' – could the authors please clarify the timing of the MRI scans more clearly (ideally with a schematic). Essentially – how many timepoints were there and what was the timing of these relative to the delivery of the tracer? Please clarify how long each scan was and how many were taken and clarify whether the patients left the scanner and were rescanned 2 hours later and what the scanning protocol was 2 hours later. If the patients left the scanner after the first scan what were their movements? as the tracer distribution may be significantly affected if they were upright vs if they lied down for example.

- Intro: 'we investigated features of early CSF tracer propagation' – not sure what you mean by 'early' as this more related to the time of the measurements rather than the physiology under investigation as the point of tracer delivery (lower spine) is not the point of CSF secretion. Perhaps you could re-write to clarify? (ie early after tracer delivery to the CSF in the spine).

-Discussion: 'From before, it has been shown that arterial hypertension, by its infliction on reduced arterial wall pulsatility, causes reduced periarterial tracer movement' – this was a very specific acute pharmacological model of hypertension where reduced wall motion was observed. To my understanding the relationship between clinical hypertension and brain vascular pulseatility is less clear with some instances of increased pulseatility in clinical hypertension. Can the authors give a more detailed discussion of the relationship between hypertension and vascular pulseatility?

- Data availability when the authors say 'Source Data are provided with this paper' what exactly does this mean? I could not see that the imaging data was available.

Reviewer #2 (Remarks to the Author):

This study is based on an analysis of a total of 75 patients that received an intrathecal administration of the contrast agent, gadobutrol. MRI was used to study dynamic fluxes of CSF tagged with gadobutrol in the subarachnoid space. The authors find that gadobutrol quickly enters along the perivascular spaces surrounding the large cerebral vessels instead of mixing with the large arachnoid space surrounding the brain. Image analysis revealed that parenchymal contrast agent influx correlated with the enrichment of contrast agent in the nearby perivascular spaces.

The authors conclude based on this extensive data set that subarachnoid space is compartmentalized to facilitate influx of CSF along the major cerebral arteries. This is an important novel observation because the CSF field presently believes that the subarachnoid space is one large compartment. The authors also report that the compartmentalization of CSF influx is compromised in aging and in patients suffering from normal pressure hydrocephalus (NPH).

The submission is based on an analysis of a unique data set that likely cannot be collected at other institutions. The author team has a long track record in producing novel and highly cited reports on CSF transport in the human brain. The current study adds new aspects to the current understanding of CSF: An effective brain waste removal system requires polarized CSF flow patterns as demonstrated by the directional influx along the periarterial spaces. Conversely, the loss of directional CSF fluid flow in aging and NPH will recirculate CSF containing amyloid and tau back into the brain. The concept of directional CSF flow is my knowledge entirely novel and its potential implication for the etiology of proteinopathies will give rise to many additional studies.

Major critique:

- The authors should be recommended for the discussion that objectively evaluates how their data can support opposing mechanisms of CSF flow.
- 'Since perivascular tracer enrichment was typical for the large artery trunks ACA, MCA and PCA, we further asked for in vivo evidence of a perivenous subarachnoid space. As shown in Supplementary Figure 4, a perivenous pattern of tracer enhancement was occasionally detected'. Question: Were these veins crossing arteries such that shunting from the periarterial to the perivenous space might have occurred?
- 'There was a considerable inter-individual variation regarding rate of antegrade perivascular transport'. Question: is it possible by looking at the physiological measurements, such as heart or respiratory rate to predict whether some of the subjects were asleep during the scanning?

- Fig. 1 is very clear and the schematic drawings help with orientation. Does the imaging have sufficient resolution to reconstruct the pial artery in 3D similar to one of the drawings?
- Were all periarterial spaces round donut shaped or where more elongated or asymmetric periarteral spaces also observed?
- Several of the panels in Fig. 5 are remarkable and could be used in textbooks. It would instructive to add drawing with labels to clearly indicate the anatomical structures.
- The publications would be a classic with a higher number of citations if the authors could include a sketch that explain CSF fluxes from the basal cisterns.

Manuscript - Response to reviewers with reference to manuscript with changes highlighted

Reviewer #1:

General comment #1:

This is an impressive study that presents a large volume of CSF-contrast-agent timeseries MRI data in order to better understand CSF dynamics in the human brain. The methods required to generate these type of data are technically v. challenging, and the large volume of high quality data that has been generated speaks to the world-leading specialist knowledge and experience that the authors have in these methods (ie lumbar delivery of CSF contrast agents in patients combined with dynamic MRI).

The manuscript presents new and exciting data that convincingly demonstrates novel characteristics of CSF physiology in the human brain. Specifically, to my knowledge, this data provides new evidence in support of para-arterial CSF transport in the human brain and that this is related to the degree of CSF-contrast agent delivery to the brain tissue. Thus, it appears that the exciting conclusion reported in the title of the paper is well supported by the novel data presented here.

The manuscript is well written and the authors appears to have given a good amount of care to the interpretation of a large a highly complex data set.

There are however some important issues that I believe should be addressed:

Answer: We thank the reviewer for thoughtful review of our work, and indeed appreciate the positive comments of the reviewer. The comments have been helpful in improving the work.

Specific comment #1:

- Figure 7. I think the ageing correlation is currently problematic as across the ~75 patients there are a number of different CSF-related conditions which have been diagnosed (table 1). These conditions are likely to be related to the age of the patient and so could be confounding the association with age that the authors seek to investigate. I recommend the authors consult a statistician to find the appropriate statistical method to test the effect of age on these MRI-based metrics given the underlying ~7 different CSF-related disease categories that exist within the data. The plots should also be color coded according to disease category to better communicate this important aspect of the data to the reader. Could the authors just look at correlation with age across the 14 ref subjects? I may have missed this but was spinal transit time accounted for in these measurements?

Answer: We thank the reviewer for this important comment. A statistician has been consulted, and updated analyses confirm that diagnosis is a confounder. The Results section has been updated, commenting that disease is a confounder (page 8, par 1). We have provided new color-coded plots but moved this figure to Supplementary Material). We think the overall correlation with age is of interest and should be reported, but our interpretation has been modified (page 8, para 1). Of note, spinal transit time was not a confounder for the presented correlations.

Specific comment #2:

- 'All MRI scans were visually assessed in a standardized manner by a board-certified radiologist 424 with 16 years' experience in neuroradiology (G.R.)' – It was not clear from the description of the methods whether the analysis was blinded or not. This is important especially given the subjective nature of several of the analysis approaches where blinding should be employed.

Answer: It has been stated in Methods that the assessment was not blinded (page 17, para 3), and commented on this in the Limitations section (page 14, last para). Blinding would of course be preferable but is not achievable in an imaging study like this, since all iNPH patients have enlarged ventricles, a finding which is very apparent in the MR images we used. Otherwise, the reader did

not consider patient diagnosis and assured that measurements were performed in a standardized manner in all patients to limit biasing.

Specific comment #3:

-Results: 'Furthermore, diffuse tracer enhancement within the subarachnoid space was typically preceded by a rim of tracer enrichment surrounding the arteries residing within the subarachnoid space... The early dissipation of the periarterial enhancement pattern'. It would help to generate some more quantitative data to support this qualitative observation. Could the authors draw ROIs in the PVS region and subarachnoid region and show that the first appearance of the tracer is indeed sooner in the PVS region vs subarachnoid space?

Answer: We thank the reviewer for this thoughtful comment and have now provided quantitative data substantiating the statement that enrichment of tracer occurred first in PVSAS and thereafter in nearby SAS. Results have been shown Fig 4f-g, described in Results (page 5, last para), and the methodology further detailed in Supplementary material (Suppl Fig 14).

Specific comment #4:

- Results: 'As shown in Supplementary Figure 4, a perivenous pattern of tracer enhancement was occasionally detected, but veins at the brain surface are generally harder to depict at MRI, and thus also perivenous enhancement. However, the dataset is highly suggestive that the foremost front of perivascular tracer enhancement in subarachnoid space is primarily periarterial, not perivenous' – from these two sentences, the reader is left unclear whether the lack of evidence for perivenous enhancement is reflective of methodological limitations (ie veins are harder to depict with MRI) or genuinely reflects the underlying physiology. Can the authors explain/substantiate their summations in more detail as this is a important/interesting finding.

Answer: This is an insightful comment. As in the rest of the body, also in the subarachnoid space venous anatomy has larger variations than on the arterial side. Moreover, being low pressure channels, veins are more susceptible to diameter change, or even collapse, than arteries. Finally, the venous wall is much thinner than the artery wall, making veins much harder to depict directly in the images, including in the T1-weighted volume scan we used. Should there exist a significant perivenous tracer enhancement in SAS, we would expect to regularly observe the circumferential enhancement pattern in regions of the SAS where we cannot confirm presence of an artery directly. This was not the case, and in the few subjects where enhancement occurred around a larger vein, the diffuse enhancement pattern in SAS had typically already appeared. Therefore, we have indication that the reason why we do not regularly observe perivenous enhancements is principally not due to methodological limitations, but rather evidence that periarterial propagation of CSF tracer in SAS is far most important, at least dominating over propagation along veins. These considerations have now been included in the Discussion (page 11, para 2).

Specific comment #5:

- Where possible, I think it would really help communicate the main findings of the work (eg. rapid Paraarterial CSF transport) by including videos showing the dynamic propagation of the tracer in the CSF spaces and ideally how this relates to the delivery of the tracer to the tissue.

Answer: We agree that this is a good idea. The revision includes Supplementary Movies 1-3 showing tracer distribution from the thecal sac towards the basal cisterns and the supratentorial subarachnoid spaces (Supplementary Movie 2-3), also zoomed in to indicate the periarterial tracer propagation (Supplementary Movie 1).

Specific comment #6:

- 'directional periarterial transport'– can the authors clarify/substantiate why they think that their data shows that periarterial transport is directional and their measurements do not reflect rapid

pulsatile-driven mixing/dispersion (with no net directionality) driving the contrast agent down a concentration gradient? (ie what is the evidence for antegrade transport)?

Answer: The evidence behind the statements is the time-delay before distal tracer appearance within the PVSAS. i.e., first-time appearance of tracer in PVSAS occurred early in proximal vessel and later in distal vessel. This reasoning has now been stated in Discussion (page 12, para 2). From this dataset, we may not conclude about the forces underlying tracer transport in PVSAS; this has to be explored in computational studies.

Specific comment #7:

- The intro is generally clear and well written. There have been a number of studies using gadolinium tracers in the CSF with dynamic T1 weighted MRI (many of which have been led by the authors of the current work), in the intro could the authors state more clearly/specifically how the design of the acquisition/ analysis in this work allows them to go above and beyond what is in the literature to address the aims of their work? Is what new question(s) does the design of the experiment allow to be addressed and why? (e.g more timepoints or higher spatial resolution of the images etc).

Answer: Our previous tracer studies explored enrichment in brain and meninges long after intrathecal tracer injection, typically addressing parenchymal tracer enrichment the morning after tracer injection, typically after about 24 hours. We have previously not examined specifically the pattern of how tracer within 2-3 hours after injection distributes within the subarachnoid spaces. This was the goal of the present study. These reflections have now been commented on in the Introduction (page 3, para 2).

Specific comment #8:

- For the 'intracranial pressure-volume reserve capacity' and 'in a dementia subtype' investigation of the MRI metrics (e.g 'first appearance'), I may have missed this but can the authors confirm/clarify that the spinal transit time was accounted for in their analysis?

Answer: We agree that this could possibly be a confounder. Therefore, we consulted a statistician and determined whether spinal transit time was a confounder. We did not find evidence for that. This aspect has now been commented on in the figure legend of Fig 8.

Specific comment #9:

- Figure 3. The analysis underlying the data presented in this figure is not described in the methods. Can the authors please add this and also clarify what question/hypothesis they were seeking to address with their analysis/ statistical tests employed here?

Answer: We thank the reviewer for this important comment and have now clarified this aspect in the Methods section (page 17, para 4). The research question was as follows: For a given blood vessel segment (A1, A2, pericallosal or M1, M2 or M3), what was the time of first appearance of perivascular contrast? Statistically, we examined whether first time appearance was different for the various segments of the vessel. Provided perivascular tracer transport was antegrade, tracer should first appear proximal and then distal in this order for anterior and medial cerebral arteries, respectively: A1 – A2 - Distal A2, and M1 – M2 – M3.

Specific comment #10:

- Figure 6. For the estimates of parenchymal enhancement, it is important to rule out partial volume of the CSF space within the parenchymal ROIS – did the authors consider this in their analysis?

Answer: The brain images were segmented using the Free Surfer software and tracer enrichment assessed within the segmented regions. The role of partial volume effects cannot be completely ruled out but should be expected to contribute to a very limited extent.

Specific comment #11:

- Can the authors generate quantitative data or show more images to support their observation of increased PVS volume in iNPH patients. Currently there just appears to be one representative image in support of this observation (supp Figure 9)?

Answer: We thank the reviewer for this comment and have now performed quantitative assessments of PVSAS dimensions in iNPH versus REF subjects. The results have been given in Fig. 9h, commented on in Results section (page 9, para 1) and Methods section (page 17, last para), and the method specified in Supplementary Material (Supplementary Fig. 15).

Specific comment #12:

- *Methods: 'The MRI acquisitions were repeated continuously during the first hour and then repeated two hours after intrathecal injection of 0.5 ml gadobutrol (1 mmol/ml) (Gadovist® 417, Bayer AB, 418 Sweden), which was performed by an experienced interventional neuroradiologist under fluoroscopic guidance at the lower lumbar level. Correct needle (22G) position within the lumbar subarachnoid space was confirmed by backflow of CSF. Patients were in the supine position when the MRI acquisitions were done' – could the authors please clarify the timing of the MRI scans more clearly (ideally with a schematic). Essentially – how many timepoints were there and what was the timing of these relative to the delivery of the tracer? Please clarify how long each scan was and how many were taken and clarify whether the patients left the scanner and were rescanned 2 hours later and what the scanning protocol was 2 hours later. If the patients left the scanner after the first scan what were their movements? as the tracer distribution may be significantly affected if they were upright vs if they lied down for example.*

Answer: We thank for this comment and have included information that MRI acquisitions were done continually with the patient in the scanner the first hour, and then repeated at 2 hours. Patients were supine during the entire period, which has been commented on (page 16, para 4).

Specific comment #13:

- *Intro: 'we investigated features of early CSF tracer propagation' – not sure what you mean by 'early' as this more related to the time of the measurements rather than the physiology under investigation as the point of tracer delivery (lower spine) is not the point of CSF secretion. Perhaps you could re-write to clarify? (ie early after tracer delivery to the CSF in the spine).*

Answer: We agree that the wording is unclear and have modified this (page 3, para 2). In this study, we explored tracer propagation during the first 2-3 hours, as opposed to our previous studies, examining tracer enrichment in brain tissue only over a much more prolonged time span.

Specific comment #14:

- *Discussion: 'From before, it has been shown that arterial hypertension, by its infliction on reduced arterial wall pulsatility, causes reduced periarterial tracer movement' – this was a very specific acute pharmacological model of hypertension where reduced wall motion was observed. To my understanding the relationship between clinical hypertension and brain vascular pulseatility is less clear with some instances of increased pulseatility in clinical hypertension. Can the authors give a more detailed discussion of the relationship between hypertension and vascular pulseatility?*

Answer: We thank the reviewer for helping us clarify this topic. In this study, we have addressed intracranial pulsatility, which is different from vascular pulsatility. In previous studies, we have explored how arterial pulsatility associate with intracranial pulsatility, which has now been commented on with a new reference, as requested by the reviewer (PMID: 29549286) (page 13, last para). With regard to the present observations, we suggest that elevated intracranial pulsatility due to underlying disease causes restriction on the arterial pulsatility, which in turn hampers the perivascular tracer transport.

Specific comment #15:

- Data availability when the authors say 'Source Data are provided with this paper' what exactly does this mean? I could not see that the imaging data was available.

Answer: The MR images per se cannot be provided due to patient data restrictions and regulations, but source data whereby statistics are derived, are included with the submission.

Reviewer #2:

General comment #1:

This study is based on an analysis of a total of 75 patients that received an intrathecal administration of the contrast agent, gadobutrol. MRI was used to study dynamic fluxes of CSF tagged with gadobutrol in the subarachnoid space. The authors finds that gadobutrol quickly enters along the perivascular spaces surrounding the large cerebral vessels instead of mixing with the large arachnoid space surrounding the brain. Image analysis revealed that parenchymal contrast agent influx correlated with the enrichment of contrast agent in the nearby perivascular spaces.

The authors conclude based on this extensive data set that subarachnoid space is compartmentalized to facilitate influx of CSF along the major cerebral arteries. This is an important novel observation because the CSF field presently believes that the subarachnoid space is one large compartment. The authors also report that the compartmentalization of CSF influx is compromised in aging and in patients suffering from normal pressure hydrocephalus (NPH).

The submission is based on an analysis of a unique data set that likely cannot be collected at other institutions. The author team has a long track record in producing novel and highly cited reports on CSF transport in the human brain. The current study adds new aspects to the current understanding of CSF: An effective brain waste removal system requires polarized CSF flow patterns as demonstrated by the directional influx along the periarterial spaces. Conversely, the loss of directional CSF fluid flow in aging and NPH will recirculate CSF containing amyloid and tau back into the brain. To concept of directional CSF flow is my knowledge entirely novel and its potential implication for the etiology of proteinopathies will give rise to many additional studies.

Answer: We thank the referee for this positive feedback and for thorough review of our work. The specific comments are addressed below.

Specific comment #1:

Major critique:

- *The authors should be recommended for the discussion that objectively evaluates how their data can support opposing mechanisms of CSF flow.*

Answer: We thank the reviewer for this positive comment, and indeed aimed as being objective in the discussion.

Specific comment #2:

• *'Since perivascular tracer enrichment was typical for the large artery trunks ACA, MCA and PCA, we further asked for in vivo evidence of a perivenous subarachnoid space. As shown in Supplementary Figure 4, a perivenous pattern of tracer enhancement was occasionally detected'. Question: Were these veins crossing arteries such that shunting from the periarterial to the perivenous space might have occurred?*

Answer: In the first place, perivenous tracer enrichment was rarely identified, as now commented in the Discussion (page 11, para 2). We found no cases where a local crossing phenomenon of arteries and veins was considered to explain occasional perivenous enhancement. Occasional perivenous enhancement was typically seen in conjunction with the appearance of a more diffuse tracer enhancement in SAS outside the PVSAS. Therefore, the evidence principally points towards a periarterial propagation of solutes in SAS, not perivenous (page 5, last para).

Specific comment #3:

• *‘There was a considerable inter-individual variation regarding rate of antegrade perivascular transport’. Question: is it possible by looking at the physiological measurements, such as heart or respiratory rate to predict whether some of the subjects were asleep during the scanning?*

Answer: We thank the reviewer for this comment. The present MRI acquisitions were obtained during the first 2-3 hours after intrathecal contrast injection. The patient stayed in the MRI lab, and during MRI scanning there was regular contact between patient and MRI technician. Between each scans lasting about 5 minutes, there was always a short oral communication between the patient and the MRI technician running the exam. For sure, no patients who went to sleep. We agree that this is important information and have now stated that participants were awake during MRI scanning (page 16, para 4).

Specific comment #4:

• *Fig. 1 is very clear and the schematic drawings help with orientation. Does the imaging have sufficient resolution to reconstruct the pial artery in 3D similar to one of the drawings?*

Answer: We thank the reviewer for this comment. The resolution is inferior for sufficient depiction of pial arteries at the brain surface, and the aim of the study was also to assess CSF tracer distribution in the subarachnoid space, including with relation to subarachnoid vessels. We have extended the set of illustrations, and provided new figures and videos (Supplementary Movie 1) to better visualize the phenomenon.

Specific comment #5:

• *Were all periarterial spaces round donut shaped or where more elongated or asymmetric periarterial spaces also observed?*

Answer: Thanks for this comment, this seemed to depend on underlying condition, and has now been commented on regarding differences between iNPH and REF (page 9, para 1).

Specific comment #6:

• *Several of the panels in Fig. 5 are remarkable and could be used in textbooks. It would instructive to add drawing with labels to clearly indicate the anatomical structures.*

Answer: This is a good comment; we have now included labels to indicate structures (Figure 5). We have also included more figures (Suppl Fig 8) and movies (Suppl Movie 2-3) to visualize the tracer enrichment over time in the basal cisterns.

Specific comment #7:

• *The publications would be a classic with a higher number of citations if the authors could include a sketch that explain CSF fluxes from the basal cisterns.*

Answer: We agree that this would be useful and have added a figure depicting the direction of CSF from below and towards the PVSAS. Figure 5 has been modified to show this, in addition to Suppl Fig 8 and Suppl Movie 2-3.

REVIEWER COMMENTS

Reviewer #1 (Remarks to the Author):

Overall, the authors have done a good job of addressing my previous concerns. Please find below some additional comments/ suggestions that I believe should be addressed prior to publication:

Major Comments:

'These observations suggest that the periarterial route may facilitate directional molecular transport within the subarachnoid space antegrade along arteries, as illustrated in Figure 3h'.

To my understanding the observation of the incremental detection of the first time appearance of the tracer down the segments of the artery (e.g. MCA M1-M3) could still be largely explained by rapid pulsatile mixing moving the tracer down a concentration gradient rather than prevailing antegrade directional transport (e.g see Figure 9 in Mechanisms of fluid movement into, through and out of the brain: evaluation of the evidence | Fluids and Barriers of the CNS | Full Text (biomedcentral.com)). For example, I have a feeling that if the tracer was injected into MCA segment M3, it would be detected a few minutes later in M2&M1 which by the authors logic would provide evidence for retrograde transport.

Whilst I agree that, intuitively the images presented here do provide tantalising evidence for directional antegrade PVS transport, to my understanding, I don't think the authors have the evidence to support this. It would be great if the authors could use their data to indeed show this was the case, but to my understanding they do not currently present evidence to support this conclusion. Irrespective, their data still clearly shows the perivascular space around the arteries to be a functionally important preferential pathway for CSF transport towards the tissue.

Thus, unless they can address the need for evidence to support their interpretation of their data demonstrating strong directional antegrade transport this interpretation should be toned down and moved away from the results (e.g. Figure 3h) and more towards the discussion.

Discussion

'Increasing age may also be accompanied with impaired periarterial transport capacity' and paragraph below (lines 352-361). I am puzzled by these statements as they imply that the authors have found a correlation with age but it is my understanding that they have no evidence to support this given the confounding effects of clinical diagnosis. Can the authors consider this when it comes to the interpretation of their data in the text of the manuscript?

Minor comments:

Abstract: 'There is a significant correlation between time of first enrichment around arteries and in nearby cerebral cortex' – to my understanding the authors do not measure time of first enrichment in the cortex but rather the % tracer enrichment which is a different measurement to what is stated here.

Figure 1B – I am very unclear what Figure 1B is and how it was generated. It does not look like an MR image (why is in colour?) but the figure legend indicates that it is. Can the authors elaborate and clarify?

Table 1 and throughout. It would be helpful to clarify what your measure of 'Tracer Enrichment' is. This could be understood to be the % of the tracer delivered to the tissue I believe it is the % MRI signal change compared to baseline. Thus the authors should clearly state how this measure of 'Tracer Enrichment' is calculated in the methods.

Methods:

Can the authors clarify how spinal transit time was determined?

'MRI acquisitions were done continually with the patient in the scanner the first hour, and then repeated at 2-3 hours after i.th. injection. Patients were supine during the entire period. The patient stayed in the MRI lab, and during MRI scanning there was regular contact between patient and MRI technician' - to me this is still a bit unclear and could benefit from some additional clarification to spell out exactly what was done. Did the authors find that taking the patient out of the scanner introduced some uncertainty to their ability to track tracer progression when comparing raw MRI T1-weighted signal intensity? Some discussion of this to guide future studies would be useful.

Results:

'We identified a perivenous tracer enrichment in few instances only' - could the authors be more specific and ideally give some numbers to back this up?

'Distal perivascular tracer enrichment occurred later than proximal perivascular tracer enrichment' this (and other similar statements) should probably be supported by an appropriate statistical test (which will likely be highly significant given the paired nature of the data).

'In Fig 4g is shown' -grammar.

'There was a considerable inter-individual variation regarding rate of antegrade perivascular transport, measured as first-time appearance of tracer along the A2 and pericallosal artery branches of ACA, and along the M2 and M3 artery branches of MCA (Supplementary Fig. 141 6)- can the incorporate the spinal transit time in order to better understand this marked variation?

'In line with this assumption, we found that the tracer in the posterior fossa and basal cisterns enriched simultaneously with early enrichment within the perivascular arachnoid space of ACA' – good to clarify that this is from visual inspection of the images.

I may have missed them, but I found it hard to find the Figure legends for the supplementary videos making them hard to understand exactly what was being depicted.

It supplementary Figure 7&8 it would be helpful to highlight the vertebral and basilar artery.

Line 311: 'Our study conclude about' -grammar.

'In addition, we found that the antegrade tracer transport in subarachnoid space was slowed with increasing age;-see my previous comment, I don't think you have the evidence for this.

Line 369 ; 'was low in the waste majority of patients' -typo.

Line 403: 'On the other hand, the reader did not consider patient diagnosis' -not sure what you mean by 'the reader', should this be 'the author'?

Observation of enlarged pvs leading to reduced transport in IIH – it might be interesting to discussion this finding in the context of the recent observations from the Nedergaard group that an increase in PVS

volume facilitates a large increase in para-arterial CSF transport to the tissue in the acute phase of stroke (which unlike the data presented here was caused by the transient collapse of the vessel).

It might also be interesting to discuss the present data in the context of the sleep wake cycle dependence of glymphatic function where from the data presented in the mouse brain (ie. Sleep Drives Metabolite Clearance from the Adult Brain - PMC (nih.gov)) the degree of CSF infiltration into the tissue recorded here will be small compared to what would be anticipated if similar measures were taken when the patients were asleep (and to what extent the measures may be correlated or otherwise).

Reviewer #2 (Remarks to the Author):

The authors have responded very well and they have implemented all the points raised during the review in the revised manuscript. It is a beautiful and important study. I have no more critique.

Manuscript - R2 - Response to reviewers with reference to manuscript with changes highlighted

Reviewer #1:

General comment #1:

Overall, the authors have done a good job of addressing my previous concerns. Please find below some additional comments/ suggestions that I believe should be addressed prior to publication:

Answer: We would like to thank the reviewer for the thorough assessment of our work. The thoughtful comments have been helpful.

Specific comment #1:

'These observations suggest that the periarterial route may facilitate directional molecular transport within the subarachnoid space antegrade along arteries, as illustrated in Figure 3h'.

To my understanding the observation of the incremental detection of the first time appearance of the tracer down the segments of the artery (e.g. MCA M1-M3) could still be largely explained by rapid pulsatile mixing moving the tracer down a concentration gradient rather than prevailing antegrade directional transport (e.g see Figure 9 in Mechanisms of fluid movement into, through and out of the brain: evaluation of the evidence | Fluids and Barriers of the CNS | Full Text (biomedcentral.com)). For example, I have a feeling that if the tracer was injected into MCA segment M3, it would be detected a few minutes later in M2&M1 which by the authors logic would provide evidence for retrograde transport.

Whilst I agree that, intuitively the images presented here do provide tantalising evidence for directional antegrade PVS transport, to my understanding, I don't think the authors have the evidence to support this. It would be great if the authors could use their data to indeed show this was the case, but to my understanding they do not currently present evidence to support this conclusion. Irrespective, their data still clearly shows the perivascular space around the arteries to be a functionally important preferential pathway for CSF transport towards the tissue.

Thus, unless they can address the need for evidence to support their interpretation of their data demonstrating strong directional antegrade transport this interpretation should be toned down and moved away from the results (e.g. Figure 3h) and more towards the discussion.

Answer: We agree with the reviewer that our study cannot exclude retrograde transport in the perivascular SAS (PVSAS). However, we would like to underline that our study setup allows for the study of physiological flow (as now commented in Discussion, lines 348-352). Our tracer flows along with the natural stream of CSF after its entry in the cisterna magna, where it joins CSF exiting the fourth ventricle. According to the generally accepted concept of CSF circulation, the major fraction of CSF is produced within the ventricular compartment before its distribution throughout the subarachnoid compartment towards the brain. With intrathecal gadobutrol, we trace CSF flow in a non-manipulated compartment and observe flow parallel to arteries towards the periphery in all subjects. Should we in theory have injected tracer at level M3 (which of course is impossible in the human setting), we cannot exclude that pulsatile mixing could have led to some retrograde flow. However, level M3 is likely not a major site for CSF production, and the validity of this experience would be heavily contested. We therefore think our observations of antegrade flow are consistent and convincing (even if not excluding the possibility of a fraction of oppositely directed flow), and too important to not be reported as a result. We sincerely ask for the reviewer's understanding for this. Furthermore, in the Discussion (lines 333-340) we substantiate our interpretation: i) Tracer

appeared significantly later in pericallosal than A2 segments of ACA (Fig. 3f), which also was apparent from visual inspection of time series of MR images (Fig. 3a-e). ii) Tracer appeared significantly later in M3 than M2 segments of MCA (Fig. 4e), which also was apparent from visual inspection of time series of MR images (Fig. 4a-d). iii) At the M2 segment of the MCA after 2 hours, the tracer enrichment was significantly stronger in the perivascular subarachnoid space (PVSAS) than the surrounding subarachnoid space (SAS), suggesting that tracer transport is facilitated along arteries. Stronger tracer enrichment in PVSAS than SAS was also apparent from visual inspection (Fig 4g). iv) Shorter first-time appearance of tracer along the ACA and MCA correlated significantly with tracer enrichment in respective cortical areas (Fig 7d-I; Suppl Fig 9a-f). Taken together, these observations suggest that the transport of tracer occurs antegrade along the arteries. The forces behind tracer transport need to be explored in future studies, preferentially using computational modeling. Moreover, how the tracer would move if administered distal along the artery, obviously needs to be modelled.

Specific comment #2:

Discussion

'Increasing age may also be accompanied with impaired periarterial transport capacity' and paragraph below (lines 352-361). I am puzzled by these statements as they imply that the authors have found a correlation with age but it is my understanding that they have no evidence to support this given the confounding effects of clinical diagnosis. Can the authors consider this when it comes to the interpretation of their data in the text of the manuscript?

Answer: We agree and have modified the discussion (lines 371-373).

Specific comment #3:

Minor comments:

Abstract: 'There is a significant correlation between time of first enrichment around arteries and in nearby cerebral cortex' – to my understanding the authors do not measure time of first enrichment in the cortex but rather the % tracer enrichment which is a different measurement to what is stated here.

Answer: Thanks! The sentence in the Abstract has been modified.

Specific comment #4:

Figure 1B – I am very unclear what Figure 1B is and how it was generated. It does not look like an MR image (why is in colour?) but the figure legend indicates that it is. Can the authors elaborate and clarify?

Answer: Figure legend for Figure 1 has been updated.

Specific comment #5:

Table 1 and throughout. It would be helpful to clarify what your measure of 'Tracer Enrichment' is. This could be understood to be the % of the tracer delivered to the tissue I believe it is the % MRI signal change compared to baseline. Thus the authors should clearly state how this measure of 'Tracer Enrichment' is calculated in the methods.

Answer: Thanks! The legend to Table 1 has been updated.

Specific comment #6:

Methods:

Can the authors clarify how spinal transit time was determined?

Answer: Thanks! The spinal transit time has now been defined in the Methods (lines 465-466).

Specific comment #7:

'MRI acquisitions were done continually with the patient in the scanner the first hour, and then repeated at 2-3 hours after i.th. injection. Patients were supine during the entire period. The patient stayed in the MRI lab, and during MRI scanning there was regular contact between patient and MRI technician' - to me this is still a bit unclear and could benefit from some additional clarification to spell out exactly what was done. Did the authors find that taking the patient out of the scanner introduced some uncertainty to their ability to track tracer progression when comparing raw MRI T1-weighted signal intensity? Some discussion of this to guide futures studies would be useful.

Answer: As already commented on, patients stayed supine in bed between scans (line 472-473).

Specific comment #8:

Results:

'We identified a perivenous tracer enrichment in few instances only' -could the authors be more specific and ideally give some numbers to back this up?

Answer: We understand the reviewer's concern about this phrasing, and we would ideally have wanted to be more precise. However, a larger precision level provided by numbers would be artificial, since the possible perivenous enhancement is much more uncertain, diffuse and sporadic, down to very few cases. Our intention was merely to give the reader some impression that perivenous enhancement is rare (and uncertain when possibly observed), and that the periarterial enhancement pattern really dominates. During our rethinking of this, we discovered one sentence where one precision could be made, and to the sentence "We identified a perivenous tracer enrichment in few instances only", we added "..., and in those cases always in conjunction with periarterial enhancement (line 107)." We also changed the wording from "few" to "very few" [instances] (line 106).

Specific comment #9:

'Distal perivascular tracer enrichment occurred later than proximal perivascular tracer enrichment' this (and other similar statements) should probably be supported by an appropriate statistical test (which will likely be highly significant given the paired nature of the data).

Answer: These results already have been shown (Figure 3f; Figure 4e; Figure 4f).

Specific comment #10:

'In Fig 4g is shown' -grammar.

Answer: Has been corrected.

Specific comment #11:

'There was a considerable inter-individual variation regarding rate of antegrade perivascular transport, measured as first-time appearance of tracer along the A2 and pericallosal artery branches of ACA, and along the M2 and M3 artery branches of MCA (Supplementary Fig. 141 6)- can the incorporate the spinal transit time in order to better understand this marked variation?

Answer: We already stated that spinal transit time was no confounder for first time appearance of tracer along the ACA and MCA (lines 138-139).

Specific comment #12:

'In line with this assumption, we found that the tracer in the posterior fossa and basal cisterns enriched simultaneously with early enrichment within the perivascular arachnoid space of ACA' - good to clarify that this is from visual inspection of the images.

Answer: Agree, this has been clarified (line 150).

Specific comment #13:

I may have missed them, but I found it hard to find the Figure legends for the supplementary videos making them hard to understand exactly what was being depicted.

Answer: Legends to movies are shown in Supplementary Material after Supplementary figures.

Specific comment #14:

It supplementary Figure 7&8 it would be helpful to highlight the vertebral and basilar artery.

Answer: The figures have been updated, and figure legends modified.

Specific comment #15:

Line 311: 'Our study conclude about' -grammar.

Answer: Has been corrected.

Specific comment #16:

'In addition, we found that the antegrade tracer transport in subarachnoid space was slowed with increasing age;-see my previous comment, I don't think you have the evidence for this.

Answer: We agree; the sentence has been removed (lines 360-361).

Specific comment #17:

Line 369 ; 'was low in the waste majority of patients' -typo.

Answer: Has been corrected.

Specific comment #18:

Line 403: 'On the other hand, the reader did not consider patient diagnosis' -not sure what you mean by 'the reader', should this be 'the author'?

Answer: This aspect has been clarified (line 418-419).

Specific comment #19:

Observation of enlarged pvs leading to reduced transport in IIH – it might be interesting to discussion this finding in the context of the recent observations from the Nedergaard group that an increase in PVS volume facilitates a large increase in para-arterial CSF transport to the tissue in the acute phase of stroke (which unlike the data presented here was caused by the transient collapse of the vessel).

Answer: While this is an interesting aspect, we here study perivascular transport in the subarachnoid spaces, while the Nedergaard group do so in the perivascular spaces of subpial arteries, which is much more peripheral and closer to the brain surface. Our observations may therefore not be transferrable to pial arteries in the stroke situation. This complicates a discussion and may be too speculative. We therefore hesitate to include a comparison of our observations in chronic iNPH and acute ischemic stroke.

Specific comment #20:

It might also be interesting to discuss the present data in the context of the sleep wake cycle dependence of glymphatic function where from the data presented in the mouse brain (ie. Sleep Drives Metabolite Clearance from the Adult Brain - PMC (nih.gov)) the degree of CSF infiltration into the tissue recorded here will be small compared to what would be anticipated if similar measures were taken when the patients were asleep (and to what extent the measures may be correlated or otherwise).

Answer: Thank you for the comment about sleep. In this study, all our patients were awake. We have previously shown that 24 hours of sleep deprivation is associated with reduced tracer clearance from brain (Eide et al. Brain 2021). Even if significant, the effect size was modest, and tracer enhancement and clearance from the human brain also occurs in the absence of sleep. We

have therefore little reason to believe that sleep would have substantially altered our observations. Of course, this should be not just be anticipated, but studied in a future, different study setup.

Reviewer #2:

General comment #1:

The authors have responded very well and they have implemented all the points raised during the review in the revised manuscript. It is a beautiful and important study. I have no more critique.

Answer: We greatly appreciate the reviewer's through review of our work.

REVIEWERS' COMMENTS

Reviewer #1 (Remarks to the Author):

In regards to the discussion around Specific Comment 1, I completely agree with the authors that the experimental findings described in the discussion of lines 333-340 provides very strong evidence that 'these observations suggest that the transport of tracer occurs antegrade along the arteries'.

However all these experimental findings (described in lines 333-340) could still be explained by rapid pulsatile back and forth flow in the PVS with no net bulk flow (since this would act to move the MRI tracer down a concentration gradient giving the impression of directional bulk flow). So the current Figure 3h would be correct if it referred to the movement of the MRI tracer but if it is referring to the underlying CSF (or 'molecular' if not referring to the tracer) then the authors are not able to say whether there is a meaningful directional bulk flow component to their observations because the measurements are not able to distinguish this.

The arguments that the authors make about the 'the generally accepted concept of CSF circulation' make sense as to why there may be an important antegrade bulk flow component in the pararterial space but in the present study the authors are taking steps into the unknown and should let these breakthrough measurements stand on their own without over interpretation based on prior assumptions. I would urge the authors not to close off another possible interpretation of the underlying fluid mechanisms that could account for their experimental data. I think it would be fine for the authors to make their case as to why they believe their data supports directional (or bulk) antegrade CSF (or molecular) transport in the paravascular space in the discussion, but I would urge caution when describing this as a 'result' as is given the impression in Figure 3h. In fact I think it is just Figure 3h which I would amend as the wording around this issue in the rest of the manuscript seems appropriately qualified.

Having re-read the manuscript I have a few further suggestions to improve clarity to the reader:

It states in the final line of the first paragraph of the discussion that 'Increasing age may also be accompanied with impaired periarterial transport capacity.' I still have reservations about this statement because of the absence of evidence to support this.

Discussion: 'While the study setup does not exclude CSF flow in the retrograde (peripheral to proximal) direction, an intrathecally injected tracer will in the cisterna magna blend with ventricular CSF and follow the physiological flow patterns within the intracranial compartment, which has by no means been

experimentally manipulated.’ – I am unclear exactly what the authors are trying to get across here, could they please rephrase/elaborate?

In the results the statement: ‘The early dissipation of the periarterial enhancement pattern suggests the artery is ensheathed by a membrane that is semipermeable to the tracer (molecular weight 695 Da; Fig. 1c-d).’ This is a potentially very interesting observation of their data (ie early dissipation) leading to an important interpretation (ie semipermeable membrane) however I think it would help if it were clarified exactly what the authors mean by early dissipation (and what data this is based on) and why this leads them to conclude that the barrier is semi-permeable.

Manuscript - R3 - Response to reviewers with reference to manuscript with changes highlighted

Reviewer #1:

Specific comment #1:

'In regards to the discussion around Specific Comment 1, I completely agree with the authors that the experimental findings described in the discussion of lines 333-340 provides very strong evidence that 'these observations suggest that the transport of tracer occurs antegrade along the arteries'.

However all these experimental findings (described in lines 333-340) could still be explained by rapid pulsatile back and forth flow in the PVS with no net bulk flow (since this would act the move the MRI tracer down a concentration gradient giving the impression of directional bulk flow). So the current Figure 3h would be correct if it referred to the movement of the MRI tracer but if it is referring to the underlying CSF (or 'molecular' if not referring to the tracer) then the authors are not able to say whether there is a meaningful directional bulk flow component to their observations because the measurements are not able to distinguish this."

Answer: We acknowledge the insightful comments of the reviewer and would like to thank for the thoughtful comments. The legend of Figure 3 has been modified, now referring to "tracer", not "molecular", transport". Also, the wording in Abstract (line 37), Results (lines 117 and 130) has been changed.

Specific comment #2:

The arguments that the authors make about the 'the generally accepted concept of CSF circulation' make sense as to why there may be an important antegrade bulk flow component in the pararterial space but in the present study the authors are taking steps into the unknown and should let these breakthrough measurements stand on their own without over interpretation based on prior assumptions. I would urge the authors not to close off another possible interpretation of the underlying fluid mechanisms that could account for their experimental data. I think it would be fine for the authors to make their case as to why they believe their data supports directional (or bulk) antegrade CSF (or molecular) transport in the paravascular space in the discussion, but I would urge caution when describing this as a 'result' as is given the impression in Figure 3h. In fact I think it is just Figure 3h which I would amend as the wording around this issue in the rest of the manuscript seems appropriately qualified.

Answer: We agree; changes have been made as described above.

Specific comment #3:

Having re-read the manuscript I have a few further suggestions to improve clarity to the reader:

It states in the final line of the first paragraph of the discussion that 'Increasing age may also be accompanied with impaired periarterial transport capacity.' I still have reservations about this statement because of the absence of evidence to support this.

Answer: To enhance the precision level, we have now omitted this sentence (lines 268-269). The Conclusion section has also been modified (line 444).

Specific comment #4:

Discussion: 'While the study setup does not exclude CSF flow in the retrograde (peripheral to proximal) direction, an intrathecally injected tracer will in the cisterna magna blend with ventricular CSF and follow the physiological flow patterns within the intracranial compartment, which has by no means

been experimentally manipulated.’ – I am unclear exactly what the authors are trying to get across here, could they please rephrase/elaborate?

Answer: Thanks for helping us clarify this; the sentence has been modified (lines 354-357).

Specific comment #5:

In the results the statement: ‘The early dissipation of the periarterial enhancement pattern suggests the artery is ensheathed by a membrane that is semipermeable to the tracer (molecular weight 695 Da; Fig. 1c-d).’ This is a potentially very interesting observation of their data (ie early dissipation) leading to an important interpretation (ie semipermeable membrane) however I think it would help if it were clarified exactly what the authors mean by early dissipation (and what data this is based on) and why this leads them to conclude that the barrier is semi-permeable.

Answer: We agree, this statement has been clarified and moved from lines 93-95 to 136-141.